# Cascaded Learned Bloom filter for Optimal Model-Filter Size Balance and Fast Rejection

## Abstract

Recent studies have demonstrated that learned Bloom filters, which combine machine learning with the classical Bloom filter, can achieve superior memory efficiency. However, existing learned Bloom filters face two critical unresolved challenges: the balance between the machine learning model size and the Bloom filter size is not optimal, and the reject time cannot be minimized effectively. We propose the Cascaded Learned Bloom Filter (CLBF) to address these issues. Our optimization approach based on dynamic programming automatically selects configurations that achieve an optimal balance between the model and filter sizes while minimizing reject time. Experiments with real-world datasets show that CLBF reduces memory usage by up to 24% and decreases reject time by up to 14 times compared to the state-of-the-art learned Bloom filter.

## 1 Introduction

Bloom filter (Bloom, 1970) is a ubiquitous data structure for approximate membership queries. By compressing a set $\mathcal{S}$ into a bit array, the Bloom filter can determine whether a query $q$ belongs to the set $\mathcal{S}$, with a small probability of false positives. Due to its high memory efficiency and fast query performance, Bloom filters have been employed in various memory-constrained and/or latency-sensitive applications (Broder & Mitzenmacher, 2004; Chang et al., 2008). Recently, a new class of Bloom filter called Learned Bloom Filter (LBF) has been proposed (Kraska et al., 2018). LBFs leverage a machine learning model that predicts whether the input belongs to the set $\mathcal{S}$ and achieves a superior memory efficiency compared to traditional Bloom filters. Despite numerous attempts to further improve memory efficiency (Mitzenmacher, 2018; Dai & Shrivastava, 2020), existing LBFs still face two critical unresolved issues: (1) the balance between the machine learning model size and the Bloom filter size is not optimal, and (2) the reject time cannot be minimized effectively.

**(1) The Balance between Machine Learning Model Size and Bloom Filter Size**: Existing LBFs lack mechanisms to automatically adjust the balance between the machine learning model and Bloom filter sizes. LBFs consist of a machine learning model and one or more Bloom filters, aiming to minimize the total memory usage, i.e., the sum of memory consumed by the machine learning model and the Bloom filters. While higher model accuracy can reduce the memory for Bloom filters, optimizing the model size is challenging due to the complex relationship between model size and accuracy. Even when increased model size improves accuracy, the balance between increased model size and reduced Bloom filter size must be evaluated. Current LBFs focus solely on optimizing Bloom filter configurations for a fixed model, neglecting model size optimization.

**(2) The Reject Time**: Existing approaches do not provide automatic methods for minimizing reject time in LBFs. The time Bloom filter takes to *reject*, i.e., answer "$q \notin S$," is often more important than the time it takes to *accept*, i.e., answer "$q \in S$." This is because the accept response may be a false positive, so a heavy verification process is often required afterward, whereas the reject response is always correct, so such a verification process is not required. The reject time is more likely to be a bottleneck than the accept time, so it is important to reduce the reject time of LBF. However, the prior research on LBFs (Fumagalli et al., 2022; Malchiodi et al., 2024) provides only heuristic guidelines to reduce reject time, and does not propose any automatic method for minimizing reject time.

To address these issues, we propose a novel learned Bloom filter with a cascaded structure called Cascaded Learned Bloom Filter (CLBF). CLFB has two key features: (1) CLBF achieves an optimal

memory size balance by training a larger machine learning model and then appropriately reducing it. (2) CLBF enables fast rejections through branching based on tentative outputs of the machine learning model and the insertion of intermediate Bloom filters. We can set the hyperparameters to control the trade-off between memory efficiency and reject time. Based on dynamic programming, our optimization approach automatically adjusts the CLBF to the optimal configuration for the given hyperparameter. Our experiments demonstrate that (1) CLBF reduces memory usage by up to 24% compared to the Partitioned Learned Bloom Filter (PLBF) (Vaidya et al., 2021), the state-of-the-art LBF for memory efficiency, and (2) CLBF reduces reject time by up to 14 times compared to PLBF.

## 2 RELATED WORK

Bloom filter (Bloom, 1970) is one of the most fundamental and widely used data structures for approximate membership queries. Bloom filters can quickly respond to queries by only performing a few hash function evaluations and checking a few bits. Although Bloom filters may return false positives, they never yield false negatives. This property makes Bloom filters valuable in memory-constrained and latency-sensitive scenarios such as networks (Broder & Mitzenmacher, 2004; Tarkoma et al., 2011; Geravand & Ahmadi, 2013) and databases (Chang et al., 2008; Goodrich & Mitzenmacher, 2011; Lu et al., 2012). The theoretical lower bound on the number of bits required for data structures supporting approximate membership queries is known to be $n \log_2(1/\varepsilon)$, where $n$ is the number of elements in the set, and $\varepsilon$ is the false positive rate (Pagh et al., 2005). Bloom filters use $\log_2(e) \cdot n \log_2(1/\varepsilon)$ bits, i.e., they require $\log_2(e)$ times more memory than the theoretical lower bound. Several improved versions of the Bloom filter, such as the Cuckoo filter (Fan et al., 2014), Vacuum filter (Wang & Zhou, 2019), Xor filter (Graf & Lemire, 2020), and Ribbon filter (Dillinger & Walzer, 2021), have been proposed to get closer to the theoretical lower bound.

Recently, the concept of Learned Bloom Filter (LBF), which further enhances the memory efficiency of Bloom filters using machine learning models, was introduced (Kraska et al., 2018). They proposed an LBF that uses a machine learning model, which predicts whether the input is included in the set $\mathcal{S}$, as a pre-filter before a classical Bloom filter (Figure 1(a)). In this LBF, elements predicted by the model as included in the set $\mathcal{S}$ are not inserted into the classical Bloom filter, while those predicted as not included are. When this LBF answers a query, it immediately answers $q \in \mathcal{S}$ if the model predicts the query is in the set. In contrast, if the model predicts the query is not in the set, this LBF uses the classical Bloom filter. This design reduces the number of elements stored in the Bloom filter, thus reducing the total memory usage. Numerous subsequent studies have sought to improve this structure further. For example, the sandwiched LBF (Mitzenmacher, 2018) sandwiches the machine learning model with two Bloom filters (Figure 1(b)). it is demonstrated that the memory efficiency is further improved by optimizing the size of the two Bloom filters. Ada-BF (Dai & Shrivastava, 2020) and the Partitioned Learned Bloom Filter (PLBF) (Vaidya et al., 2021) further enhance memory efficiency by utilizing the *score*, which is the prediction of the machine learning model regarding the likelihood that an input element is included in the set (Figure 1(c)). These LBFs employ multiple Bloom filters with different false positive rates, selecting the appropriate filter based on the score. This approach allows for a more continuous and fine-grained utilization of the model predictions.

While most research on LBFs has focused on the optimal configuration of Bloom filters for a fixed trained machine learning model, some studies have investigated the choice of the machine learning model itself (Fumagalli et al., 2022; Dai et al., 2022; Malchiodi et al., 2024). These studies evaluate various machine learning models and LBF configuration methods (such as sandwiched LBF and PLBF) across different datasets, measuring memory efficiency and reject times. The results suggest that the optimal machine learning model varies depending on dataset noisiness, ease of learning, and the importance of minimizing reject time. However, these studies provide only general guidelines based on observed trends, and no method has yet been proposed for automatically selecting the optimal machine learning model.

Optimizing the hash functions is another approach to improving accuracy using key and non-key data. Hash Adaptive Bloom Filter (HABF) (Xie et al., 2021b) uses a lightweight data structure called HashExpressor to select suitable hash functions for each key, and Projection Hash Bloom Filter (PHBF) (Bhattacharya et al., 2022a) employs projections as hash functions. Unlike LBFs, these approaches avoid classifier training and instead pack information into a lighter structure.

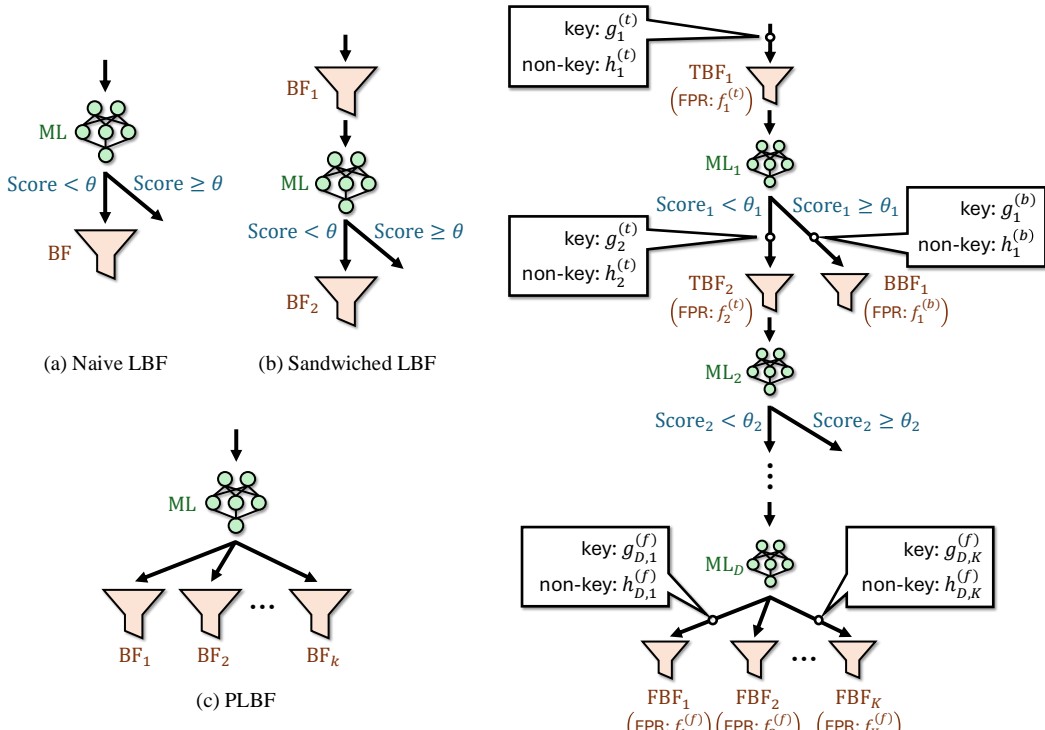

Figure 1: The architecture of Existing LBFs: (a) Naive LBF (Kraska et al., 2018) has a single backup Bloom filter. (b) Sandwiched LBF (Mitzenmacher, 2018) applies a pre-filter before the model inference. (c) PLBF (Vaidya et al., 2021) uses multiple backup Bloom filters.

Figure 2: The architecture of CLBF: CLBF alternates between score-based branching and Bloom filter-based filtering. This design generalizes the architectures of sandwiched LBF and PLBF. Note that $g_*^{(*)}$ and $h_*^{(*)}$ represent the proportions of keys and non-keys passing through each root when **filtering using TBFs is not performed**.

## 3 METHOD

This section describes the architecture and construction method of our proposed Cascaded Learned Bloom Filter (CLBF). In Section 3.1, we describe the architecture of CLBF. Then, we formulate the problem of constructing CLBF in Section 3.2, and the optimization method for configuring CLBF using dynamic programming is described in Section 3.3.

### 3.1 ARCHITECTURE OVERVIEW

CLBF employs a cascade structure consisting of multiple machine learning models and multiple Bloom filters arranged alternately (Figure 2). The structure of CLBF, where models are aligned sequentially, is similar to ensemble learning techniques such as boosting (Freund & Schapire, 1997; Friedman, 2001). In fact, the machine learning models used in our experiments (Section 4) are weak learners trained using a boosting algorithm. By automatically determining the appropriate number of machine learning models (weak learners) and adjusting the false positive rates of each Bloom filter, CLBF not only balances model and filter sizes but also shortens the reject time (as detailed in Section 3.2 and Section 3.3).

Now, we explain how the CLBF processes a query. A query $q$ first enters the initial *Trunk Bloom filter*, $TBF_1$. If $TBF_1$ determines that $q \notin \mathcal{S}$, CLBF immediately concludes $q \notin \mathcal{S}$. Otherwise, the query is passed to the first machine learning model, $ML_1$, which outputs a score indicating how likely $q$ is included in $\mathcal{S}$. If this score exceeds a threshold $\theta_1$, a subsequent *Branch Bloom filter*, $BBF_1$, is used to make the final determination. Otherwise, $q$ is passed to the next Trunk Bloom filter, $TBF_2$. In this way, CLBF alternates between branching based on the tentative score and filtering

using a Bloom filter. When the query reaches the $D$-th (i.e., final) machine learning model, CLBF adopts the same approach as PLBF. In other words, multiple thresholds are set, and based on the range in which the query score falls, one of the $K$ *Final Bloom filters* ($\mathrm{FBF}_1, \mathrm{FBF}_2, \ldots, \mathrm{FBF}_K$) is selected for use. When inserting a key into CLBF, the key is inserted into all Bloom filters traversed by the process outlined above, ensuring the absence of false negatives. The exact algorithms for key insertion and query responses are detailed in Appendix A.

The CLBF architecture generalizes existing LBFs. CLBF with only one Trunk Bloom filter and one Final Bloom filter is equivalent to a sandwiched LBF. Moreover, CLBF that omits both Trunk and Branch Bloom filters while employing multiple Final Bloom filters corresponds to a PLBF. By selecting the optimal configuration under this generalized architecture, CLBF can achieve better memory efficiency and shorter reject time.

The notations for CLBF description are as follows (most of the variables defined here are illustrated in Figure 2): $S$ represents the set stored by CLBF, and $n$ denotes the number of elements in $S$. $Q$ refers to the set of elements not in $S$, used for construction of CLBF. The elements in the set $S$ are called *keys*, and the elements not in the set $S$ are called *non-keys*. $D$ denotes the number of machine learning models CLBF uses ($D$ is a value optimized via dynamic programming). $\mathrm{ML}_d$ ($d \in \{1, 2, \ldots, D\}$) represents the $d$-th machine learning model. $\mathrm{TBF}_d$ represents the Trunk Bloom filter used just before $\mathrm{ML}_d$, and its false positive rate is denoted by $f_d^{(t)}$. $\mathrm{BBF}_d$ ($d \in \{1, 2, \ldots, D-1\}$) represents the Branch Bloom filter used when the output of $\mathrm{ML}_d$ exceeds the threshold $\theta_d$, and its false positive rate is denoted by $f_d^{(b)}$. $\mathrm{FBF}_k$ ($k \in \{1, 2, \ldots, K\}$) denotes the $k$-th Final Bloom filter associated with the $k$-th smallest score range output by $\mathrm{ML}_D$, and its false positive rate is denoted by $f_k^{(f)}$, where $K$ is the number of Final Bloom filters. $\mathrm{Size}(\cdot)$ represents the memory size of the input (e.g., $\mathrm{Size}(\mathrm{ML}_d)$ represents the memory size of $\mathrm{ML}_d$). $\mathrm{Time}(\cdot)$ represents the inference time of the input (e.g., $\mathrm{Time}(\mathrm{ML}_d)$ represents the inference time of $\mathrm{ML}_d$).

During CLBF construction, the sets $S$ and $Q$ (or their subsets) are used as validation data to measure the proportion of keys and non-keys passed to each Bloom filter and machine learning model. In this measurement, filtering using TBFs is not performed; only the branching based on the tentative outputs of the machine learning models is applied. We define $g_d^{(t)}$ and $h_d^{(t)}$ as the proportions of keys and non-keys in the validation data that are passed to $\mathrm{TBF}_d$, respectively. Similarly, we define $g_d^{(b)}$ and $h_d^{(b)}$ as the proportions of keys and non-keys in the validation data that are passed to $\mathrm{BBF}_d$, respectively. Finally, we define $g_{D,k}^{(f)}$ and $h_{D,k}^{(f)}$ as the proportions of keys and non-keys passed to $\mathrm{FBF}_k$ when the number of machine learning models used by CLBF is $D$.

## 3.2 PROBLEM FORMULATION

The following components must be provided to construct the CLBF: several pre-trained machine learning models, two hyperparameters, and a validation dataset. The machine learning models are trained to perform binary classification between key and non-key using all or part of the sets $S$ and $Q$. The number of pre-trained models is denoted by $\bar{D}$. Among these, only the first $D$ ($\leq \bar{D}$) models are used in the CLBF, where $D$ is a parameter optimized during CLBF construction. The two hyperparameters are $F$ ($\in (0, 1)$), which represents the target false positive rate for the CLBF, and $\lambda$ ($\in [0, 1]$), which controls the trade-off between memory efficiency and reject time. The CLBF is optimized to minimize memory usage and reject time, subject to the constraint that the expected false positive rate does not exceed $F$. As $\lambda$ increases, greater emphasis is placed on memory efficiency; specifically, when $\lambda = 1$, only memory efficiency is optimized, while when $\lambda = 0$, only reject time is minimized.

The following four kinds of parameters are optimized for constructing the CLBF: (i) $D$, i.e., the depth of the CLBF, (ii) $f_d^{(t)}$ ($d \in \{1, 2, \ldots, D\}$), i.e., the false positive rate of $\mathrm{TBF}_d$, (iii) $f_d^{(b)}$ ($d \in \{1, 2, \ldots, D-1\}$), i.e., the false positive rate of $\mathrm{BBF}_d$, and (iv) $f_k^{(f)}$ ($k \in \{1, 2, \ldots, K\}$), i.e., the false positive rate of $\mathrm{FBF}_k$. Another parameter, $\theta_d$, which represents the threshold for branching based on the tentative score output by $\mathrm{ML}_d$ ($d \in \{1, \ldots, D-1\}$), should also be optimized. However, jointly optimizing $\theta_d$ with the other parameters is too challenging, as the value of $\theta_d$ affects the score distributions output by $\mathrm{ML}_{d+1}$ and subsequent models, and capturing these effects is difficult. Thus, we adopt a heuristic approach, evaluating several candidates

$\boldsymbol{\theta}$ $(= [\theta_1, \theta_2, \ldots, \theta_{\bar{D}-1}])$ and selecting the one that minimizes the objective function. Specifically, for each $\alpha$ $(\in \{0.5, 0.2, 0.1, \ldots, 0.0001, 0.0\})$, we evaluate $\boldsymbol{\theta}$ such that $\theta_d$ is the top-$\alpha$ (ratio) score of non-keys output by $\mathrm{ML}_d$. For the final machine learning model, $\mathrm{ML}_D$, the thresholds are set to maximize the KL divergence between the score distributions of keys and non-keys, following the same method as in the PLBF (Vaidya et al., 2021).

The following objective function is minimized under the constraint that the "expected" false positive rate does not exceed $F$:

$$\frac{\lambda}{M} \cdot (\text{Memory Size}) + \frac{1 - \lambda}{R} \cdot \mathbb{E}[\text{Reject Time}]. \quad (1)$$

Here, MemorySize represents the memory usage of the CLBF, and $\mathbb{E}[\text{RejectTime}]$ denotes the "expected" time to reject a non-key query. The "expected" values are calculated under the assumption that the validation data and the query are sampled from the same distribution. The constants $M$ and $R$ are scaling factors to align the units of the two terms, with $M$ representing the memory usage and $R$ representing the reject time of a standard classic Bloom filter under the same settings of $n$ and $F$. These constants are measured using the validation dataset. Each term of the objective function, Equation (1), can be written as follows:

$$\text{Memory Size} = \sum_{d=1}^{D} \text{Size}(\mathrm{ML}_d) + \sum_{d=1}^{D} \text{Size}(\mathrm{TBF}_d) + \sum_{d=1}^{D-1} \text{Size}(\mathrm{BBF}_d) + \sum_{k=1}^{K} \text{Size}(\mathrm{FBF}_k), \quad (2)$$

$$\mathbb{E}[\text{Reject Time}] = \sum_{d=1}^{D} \left( \text{Time}(\mathrm{ML}_d) \cdot h_d^{(t)} \prod_{i=1}^{d} f_i^{(t)} \right). \quad (3)$$

In the formula for the expected reject time, Equation (3), it is assumed that the reject time can be approximated by the total time taken by the inference of the machine learning models. $\text{Time}(\mathrm{ML}_d)$ is a constant obtained by performing several inference runs and averaging the observed inference times. Here, note that the factor $h_d^{(t)} \prod_{i=1}^{d} f_i^{(t)}$ in Equation (3) represents the expected proportion of non-key queries inferred by $\mathrm{ML}_d$. This expected proportion is expressed by this factor because only non-keys that have passed through all of $\mathrm{TBF}_i$ ($i \in \{1, \ldots, d\}$) as false positives are inferred by $\mathrm{ML}_d$.

### 3.3 DYNAMIC PROGRAMMING SOLUTION FOR CLBF CONSTRUCTION

Here, we introduce the dynamic programming method for finding the parameters that minimize the objective function given in Equation (1). The following is a crucial insight for the dynamic programming approach: once $D$ and $\boldsymbol{f}^{(t)}$ $(= [f_1^{(t)}, f_2^{(t)}, \ldots, f_D^{(t)}])$ are obtained, the optimal values for the remaining parameters, $\boldsymbol{f}^{(b)}$ $(= [f_1^{(b)}, f_2^{(b)}, \ldots, f_{D-1}^{(b)}])$ and $\boldsymbol{f}^{(f)}$ $(= [f_1^{(f)}, f_2^{(f)}, \ldots, f_K^{(f)}])$, can be determined immediately. This conclusion follows from two observations. First, the expected reject time, given in Equation (3), depends only on $D$ and $\boldsymbol{f}^{(t)}$. Second, once $D$ and $\boldsymbol{f}^{(t)}$ are given, the proportions of keys and non-keys entering each Bloom filter become known. Following the PLBF approach, we can optimize the false positive rates of each Bloom filter to minimize the total memory size, given in Equation (2). For a Bloom filter where the proportion of $g$ from $\mathcal{S}$ and the proportion of $h$ from $\mathcal{Q}$ are entered, it is often optimal to set the false positive rate to $Fg/h$ (more precisely, it is necessary to consider cases where this exceeds 1). Therefore, once $D$ and $\boldsymbol{f}^{(t)}$ are determined, the parameters $\boldsymbol{f}^{(b)}$ and $\boldsymbol{f}^{(f)}$ are immediately determined, so from now on we focus on optimizing $D$ and $\boldsymbol{f}^{(t)}$.

Now, we explain the dynamic programming approach to optimize $D$ and $\boldsymbol{f}^{(t)}$. In determining the false positive rate of the Bloom filters *under* $\mathrm{TBF}_d$—that is, $\mathrm{BBF}_d, \mathrm{BBF}_{d+1}, \ldots, \mathrm{BBF}_{D-1}$ and $\mathrm{FBF}_1, \mathrm{FBF}_2, \ldots, \mathrm{FBF}_K$—, the factor $\prod_{i=1}^{d-1} f_i^{(t)}$ plays a key role because the proportion of non-key queries that enter these Bloom filters is proportional to this factor. Therefore, we introduce the function $\mathrm{dp}(d, T) : \{1, 2, \ldots, \bar{D}\} \times (0, 1] \to \mathbb{R}$, which is defined intuitively (but informally) as "the minimum value of the objective function under $\mathrm{TBF}_d$, subject to the constraint that $\prod_{i=1}^{d-1} f_i^{(t)} = T$." More precisely, $\mathrm{dp}(d, T)$ is defined as follows:

$$\mathrm{dp}(d, T) := \min_{\substack{D, \boldsymbol{f}^{(t)} \\ \text{s.t. } \prod_{i=1}^{d-1} f_i^{(t)} = T}} \left( \frac{\lambda}{M} \cdot (\text{M.S. under } \mathrm{TBF}_d) + \frac{1 - \lambda}{R} \cdot (\text{R.T. under } \mathrm{TBF}_d) \right), \quad (4)$$

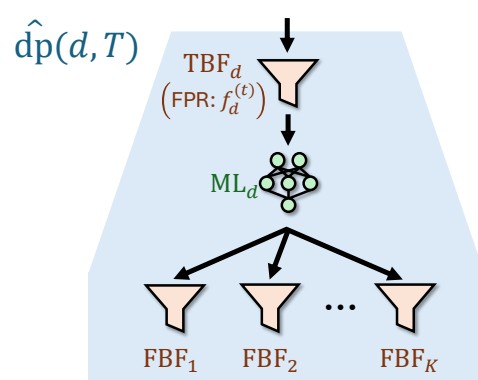
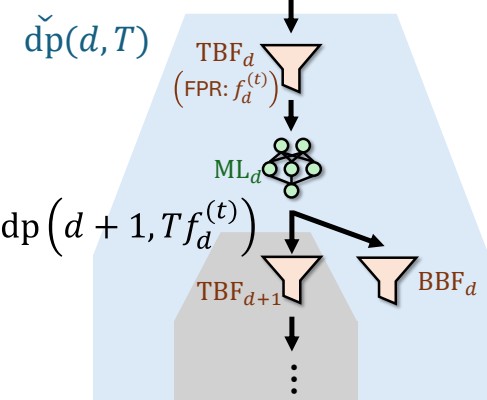

(a) $\hat{\mathrm{dp}}(d, T)$ is the minimum objective function value under $\mathrm{TBF}_d$ subject to the constraint that $D = d$.

(b) $\check{\mathrm{dp}}(d, T)$ is the minimum objective function value under $\mathrm{TBF}_d$ subject to the constraint that $D > d$.

Figure 3: The value of $\mathrm{dp}(d, T)$ is calculated by selecting the appropriate value from the case where $D = d$, i.e., $\hat{\mathrm{dp}}(d, T)$, and the case where $D > d$, i.e., $\check{\mathrm{dp}}(d, T)$. The value of $\mathrm{dp}(d + 1, T f_d^{(t)})$ is used recursively to calculate $\check{\mathrm{dp}}(d, T)$.

where (M.S. under $\mathrm{TBF}_d$) is defined as

$$\sum_{i=d}^{D} \mathrm{Size}(\mathrm{ML}_i) + \sum_{i=d}^{D} \mathrm{Size}(\mathrm{TBF}_i) + \sum_{i=d}^{D-1} \mathrm{Size}(\mathrm{BBF}_i) + \sum_{k=1}^{K} \mathrm{Size}(\mathrm{FBF}_k), \tag{5}$$

and (R.T. under $\mathrm{TBF}_d$) is defined as

$$\sum_{i=d}^{D} \left( \mathrm{Time}(\mathrm{ML}_i) \cdot h_i^{(t)} T \prod_{j=d}^{i} f_j^{(t)} \right). \tag{6}$$

Here, by substituting $d = 1, T = 1$ into Equation (4), we can see that $\mathrm{dp}(1, 1)$ is equivalent to the minimum value of Equation (1) because the constraint $\prod_{i=1}^{d-1} f_i^{(t)} = T$ is no longer in effect when $d = 1, T = 1$ and the objective function for $\min$ in Equation (4) is the same as Equation (1) when $d = 1$. In order to find $\mathrm{dp}(1, 1)$, we perform dynamic programming, setting the false positive rate of the BBFs and the FBFs to be $\min(Fg/h, 1)$, where $g$ and $h$ are the expected ratios of keys and non-keys that are input to the Bloom filter, taking into account the filtering of the TBFs.

Here, we define two functions, $\tilde{f}(g, h)$ and $\tilde{s}(g, \epsilon)$, which we use to explain how to calculate $\mathrm{dp}(d, T)$, as follows:

$$\tilde{f}(g, h) = \min\left(\frac{Fg}{h}, 1\right), \quad \tilde{s}(g, \epsilon) = cng \cdot \log_2\left(\frac{1}{\epsilon}\right). \tag{7}$$

In other words, $\tilde{f}(g, h)$ represents the false positive rate that is tentatively set when performing dynamic programming for BBF and FBF, where $g$ and $h$ are the expected ratios of keys and non-keys that are input to the Bloom filter. $\tilde{s}(g, \epsilon)$ represents the memory usage of a Bloom filter with a false positive rate of $\epsilon$ held by $n \cdot g$ keys ($c = \log_2(e)$ for the standard Bloom filter, and if a variant of the Bloom filter is used instead of the Bloom filter as TBFs, BBFs, and FBFs, then $c$ will be a different value).

Next, we explain how to calculate $\mathrm{dp}(d, T)$. We consider two cases and adopt the one that yields the better result as $\mathrm{dp}(d, T)$. The first case is when $D = d$, i.e., where the elements are distributed across the $K$ FBFs according to the output of $\mathrm{ML}_d$ (Figure 3a). Subject to the constraint that

$\prod_{i=1}^{d-1} f_i^{(t)} = T$ and $D = d$, the minimum objective function value under $\text{TBF}_d$ is

$$\hat{\text{dp}}(d,T) = \min_{f_d^{(t)} \in (0,1]} \frac{\lambda}{M} \cdot \left( \text{Size}(\text{ML}_d) + \tilde{s}(g_d^{(t)}, f_d^{(t)}) + \sum_{k=1}^{K} \tilde{s}\left( g_{d,k}^{(f)}, \tilde{f}(g_{d,k}^{(f)}, h_{d,k}^{(f)} T f_d^{(t)}) \right) \right) +$$

$$\frac{1-\lambda}{R} \cdot \text{Time}(\text{ML}_d) \cdot h_d^{(t)} T f_d^{(t)}. \quad (8)$$

The second case is when $D > d$, i.e., where the elements are distributed across $\text{TBF}_{d+1}$ and $\text{BBF}_d$ according to the output of $\text{ML}_d$ (Figure 3b). Under the condition that $\prod_{i=1}^{d-1} f_i^{(t)} = T$ and $D > d$, the minimum objective function value under $\text{TBF}_d$ can be computed using $\text{dp}(d+1, T f_d^{(t)})$ as follows:

$$\check{\text{dp}}(d,T) = \min_{f_d^{(t)} \in (0,1]} \frac{\lambda}{M} \cdot \left( \text{Size}(\text{ML}_d) + \tilde{s}(g_d^{(t)}, f_d^{(t)}) + \tilde{s}\left( g_d^{(b)}, \tilde{f}(g_d^{(b)}, h_d^{(b)} T f_i^{(t)}) \right) \right) +$$

$$\frac{1-\lambda}{R} \cdot \text{Time}(\text{ML}_d) \cdot h_d^{(t)} T f_d^{(t)} + \text{dp}(d+1, T f_d^{(t)}). \quad (9)$$

Then, we can compute $\text{dp}(d,T)$ as follows:

$$\text{dp}(d,T) = \begin{cases} \min\left( \hat{\text{dp}}(d,T), \check{\text{dp}}(d,T) \right) & d \in \{1, \ldots, \bar{D} - 1\}, \\ \hat{\text{dp}}(d,T) & d = \bar{D}. \end{cases} \quad (10)$$

Now, we explain the implementation of dynamic programming and the procedure for determining the configuration of CLBF. Although we have defined $\hat{\text{dp}}(d,T)$ and $\check{\text{dp}}(d,T)$ as the minimum values of a complex function over $f_d^{(t)} \in (0,1]$, finding the exact minimum is challenging. Therefore, we approximate the minimum by evaluating $P$ discrete values for $f_d^{(t)}$ and selecting the best result. Specifically, we evaluate $f_d^{(t)}$ over the discrete set $\{p^0, p^1, \ldots, p^{P-1}\}$. Similarly, we treat $T$—the second argument of dp—as taking values from the same set, i.e., $T \in \{p^0, p^1, \ldots, p^{P-1}\}$. By recursively calculating $\text{dp}(1,1)$, we can determine the optimal $D$ and $\boldsymbol{f}^{(t)}$. Once $D$ and $\boldsymbol{f}^{(t)}$ are fixed, the optimal $\boldsymbol{f}^{(b)}$ and $\boldsymbol{f}^{(f)}$ can be determined using the PLBF approach. Through this dynamic programming method, we can find the optimal configuration of CLBF.

Here, we show that the computational complexity of this dynamic programming is $\mathcal{O}(\bar{D}P^2 + \bar{D}PK)$. The number of possible values for $d$—the first argument of dp—is $\bar{D}$, and the number of possible values for $T$—the second argument of dp—is $P$. For each pair $(d,T)$, up to $P$ values of $f_d^{(t)}$ are considered. In the definition of $\hat{\text{dp}}(d,T)$, i.e., Equation (8), the function inside the $\min$ contains a summation from $k = 1$ to $k = K$, so if we compute it naively, $\mathcal{O}(K)$ computations are required to evaluate the function. However, by precomputing this summation for each $d$ and "$T f_d^{(t)}$," we can evaluate this function in $\mathcal{O}(1)$. The time complexity of this precomputation is $\mathcal{O}(\bar{D}PK)$, as there are $\bar{D}$ values for $d$, $P$ values for "$T f_d^{(t)}$", and $\mathcal{O}(K)$ computations are required for each case. Additionally, the function inside the $\min$ in the definition of $\check{\text{dp}}(d,T)$, i.e., Equation (9), can be evaluated in $\mathcal{O}(1)$. Therefore, the total computational complexity of this dynamic programming is $\mathcal{O}(\bar{D}P^2 + \bar{D}PK)$. This dynamic programming method is repeated for different sets of parameters $\boldsymbol{\theta}$, and the $\boldsymbol{\theta}$ that minimizes the objective function, Equation (1), is selected.

## 4 EXPERIMENTS

In this section, we evaluate the memory efficiency, reject time, and construction time of CLBF by comparing it with a standard Bloom filter and existing learned Bloom filters. The memory usage of learned Bloom filters includes the size of the machine learning model. In the following, training data refers to the data used for training the machine learning model, validation data refers to the data used for configuring the LBF, and test data refers to the data used to measure the accuracy and reject time of the Bloom filter and LBF. We conducted experiments using the following two datasets:

**Malicious URLs Dataset**: Following the previous studies on learned Bloom filters (Dai & Shrivastava, 2020; Vaidya et al., 2021), we used the Malicious URLs dataset (Siddhartha, 2021). This

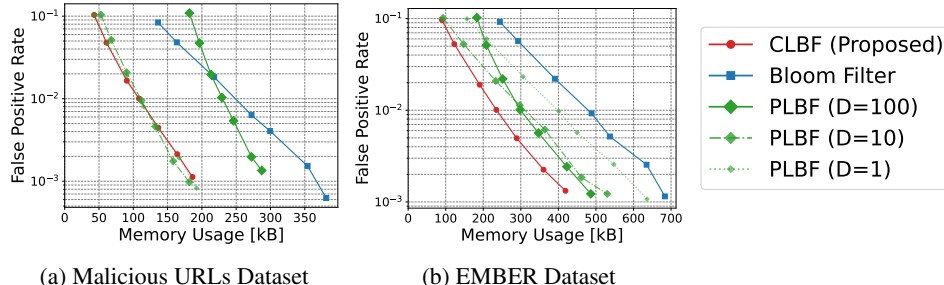

(a) Malicious URLs Dataset      (b) EMBER Dataset

Figure 4: Trade-off between memory usage and accuracy (lower-left is better): CLBF achieves equal to or better memory efficiency than any other PLBF with $D$.

dataset contains 223,088 malicious URLs and 428,103 benign URLs. The set of all malicious URLs constitutes the set $\mathcal{S}$, which the Bloom filters aim to store. We divided the benign URLs into 80% as training data, 10% as validation data, and 10% as test data. All malicious URLs were used as training data and validation data.

**EMBER Dataset**: Following the previous studies on learned Bloom filters (Vaidya et al., 2021; Sato & Matsui, 2023), we used the EMBER dataset (Anderson & Roth, 2018). This dataset contains 400,000 malicious files and 400,000 benign files, their vectorized features, and sha256 hashes (the 200,000 unlabeled files were not used in this experiment). We used 10% of the benign files as training data, 10% as validation data, and 80% as test data. For the malicious files, 10% were used as training data, while all were used as validation data. This split ratio was adopted to avoid excessive training time due to the high dimensionality of the features in the EMBER dataset.

All experiments were implemented in C++ and conducted on a Linux machine equipped with an Intel® Core™ i9-11900H CPU @ 2.50 GHz and 62 GB of memory. The code was compiled using GCC version 11.4.0 with the -O3 optimization flag, and all experiments were performed in single-threaded mode. Although any machine learning model can be used, we employed XGBoost (Chen & Guestrin, 2016), a widely used implementation of gradient boosting. Each weak learner in XGBoost corresponds to the machine learning models $\mathrm{ML}_1, \ldots, \mathrm{ML}_{\bar{D}}$ in our proposed method. Here, to clearly demonstrate the effectiveness of our CLBF, we have omitted the results of some baselines. For more extensive experimental results, please see Appendix C.

### 4.1 MEMORY AND ACCURACY

We compared the trade-off between memory usage and accuracy of CLBF with that of a standard Bloom filter and existing learned Bloom filters, specifically PLBF (Vaidya et al., 2021). Sandwiched LBF (Mitzenmacher, 2018) is omitted here, as it performs worse than PLBF in this trade-off. This trade-off is controlled by varying the hyperparameter $F$ from 0.1 to 0.001. In this evaluation, we set $\lambda = 1.0$, meaning CLBF is optimized solely for memory efficiency. The number of boosting rounds used for training XGBoost in constructing CLBF, i.e., $\bar{D}$, was set to 100. For PLBF, we present results using XGBoost with training rounds $D \in \{1, 10, 100\}$, as PLBF lacks a mechanism for automatically adjusting the model size.

Figure 4 presents the results. It shows that CLBF consistently achieves equal or better memory efficiency than PLBF, while CLBF does not need to evaluate different $D$ values as PLBF does. In the Malicious URLs dataset, PLBF achieves optimal memory efficiency with $D = 1$ or $D = 10$, while CLBF closely matches this efficiency. Specifically, the optimal $D$ selected by CLBF in the Malicious URLs dataset ranges from 1 to 18. On the other hand, in the EMBER dataset, PLBF performs best with $D = 10$ when $F > 10^{-2}$ and with $D = 100$ when $F < 10^{-2}$. Across all values of $F$, CLBF outperforms PLBF in terms of memory efficiency. The optimal $D$ chosen by CLBF in the EMBER dataset ranges from 18 to 38. At $F = 0.01$, CLBF achieves a 24% reduction in memory usage compared to PLBF with $D = 10$ and $D = 100$. These results indicate that the optimal model size $D$ for achieving the best memory efficiency varies depending on both the dataset and the value of $F$, and that our CLBF can automatically select the optimal $D$.

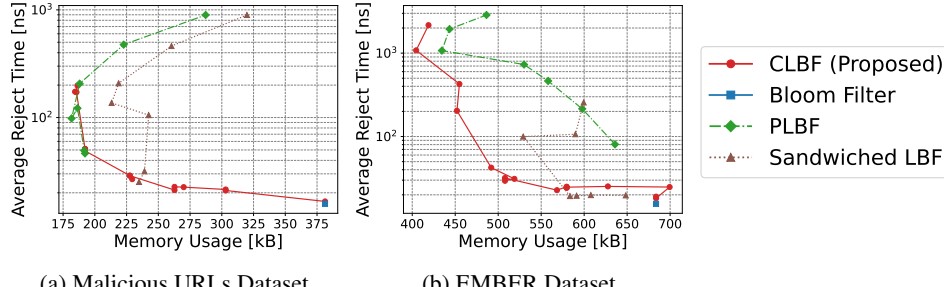

(a) Malicious URLs Dataset          (b) EMBER Dataset

Figure 5: Trade-off between memory usage and average reject time (lower-left is better): Compared to PLBF with similar memory efficiency, CLBF achieves a significantly shorter average reject time (up to 14 times shorter).

## 4.2 MEMORY AND REJECT TIME

We compared the trade-off between memory usage and reject time in CLBF against existing learned Bloom filters and the standard Bloom filter. This trade-off in CLBF is controlled by varying the hyperparameter $\lambda$ within the range $[0, 1]$. We set the false positive rate $F$ for all methods to 0.001. The number of XGBoost training rounds (corresponding to $\bar{D}$ in our method) was set to 100 for CLBF. Since the standard Bloom filter lacks a parameter to control this trade-off, it is represented as a point. For sandwiched LBF and PLBF, we observed the changes in memory usage and reject time when varying $D$ from 1 to 100.

Figure 5 illustrates the results. We can see that CLBF greatly outperforms sandwiched LBF in terms of memory efficiency, and that CLBF greatly outperforms PLBF in terms of reject time. While the reject time of sandwiched LBF tends to be shorter than that of PLBF, its memory efficiency is inferior to both PLBF and CLBF. For example, in the EMBER dataset, CLBF achieves a minimum memory usage of approximately $400\,\mathrm{kB}$, whereas no sandwiched LBF configuration uses less than $500\,\mathrm{kB}$. On the other hand, although PLBF can achieve comparable memory efficiency to CLBF, it suffers from significantly slower reject times. Specifically, when the memory usage is around $500\,\mathrm{kB}$ in the EMBER dataset, the average reject time of CLBF is approximately 14 times shorter than that of PLBF.

Additionally, we can see that the CLBF plot forms a curve that is almost a Pareto front. In most cases, no other learned or non-learned Bloom filter can achieve both lower memory usage and shorter reject time than CLBF. Moreover, for CLBF itself, improving memory efficiency (or worsening it) results in a corresponding worsening (or improvement) of reject time. By contrast, in other LBFs, both the memory efficiency and reject time worsen as $D$ increases when $D$ is too large. Since it is impossible to know this turning point for $D$ in advance, an imprudent choice of $D$ risks constructing an LBF that is inefficient in terms of both memory and reject time. Our CLBF has the advantage of avoiding the risk of setting such a needlessly large $D$.

## 4.3 CONSTRUCTION TIME

This section compares the time required to construct CLBF with other existing LBFs and the standard Bloom filter. The comparison is made with fast PLBF (Sato & Matsui, 2023), a method that constructs the same data structure as PLBF more quickly.

The results are shown in Figure 6. Here, "Scoring Time" refers to the time taken to measure the score of each sample against each machine learning model by passing validation data through them. "Configuration Time" refers to the time required to compute the optimal configuration using the results of the scoring phase. In the case of CLBF, the configuration process involves dynamic programming, as described in Section 3.3. Similarly, the configuration for sandwiched LBF and fast PLBF involves determining the optimal thresholds and false positive rates for each Bloom filter based on the scoring.

The results indicate that CLBF requires a longer configuration time than other existing LBFs. Compared to the construction time of existing LBFs with the same machine learning model size

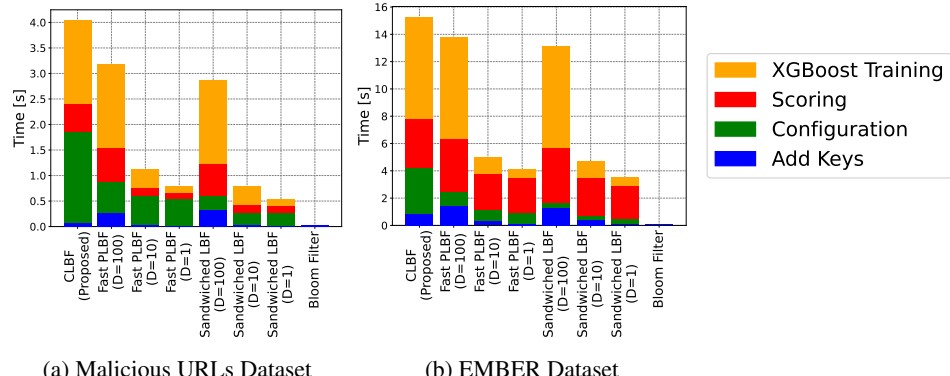

(a) Malicious URLs Dataset        (b) EMBER Dataset

Figure 6: Construction time: CLBF requires additional computation time of 10% to 41% compared to the construction time of existing LBFs using the same size machine learning model ($D = 100$).

($D = 100$), the construction time of CLBF is approximately 10% to 40% longer. However, we believe that this additional overhead is a minor drawback. Considering that the construction time for the smallest sandwiched LBF is already about 100 times longer than that of a standard Bloom filter, we can assume that LBF is not something that is used in scenarios where construction speed is sensitive. LBFs should be used in contexts where the frequency of reconstruction is low (once an hour or less). For example, the (learned) Bloom filter used to filter malicious URLs does not need to be rebuilt frequently because the set of malicious URLs does not change that quickly. In such cases, the construction time of CLBF, which is 1.4 times longer than that of the sandwiched LBF, is not a problem, and the benefits of the optimal configuration obtained by searching virtually all cases of $D \in \{1, 2, \dots, 100\}$ are considered to be greater.

## 5    LIMITATION

Although our method is highly compatible with learning models composed of multiple weak learners, such as boosting, it cannot be directly applied to models composed of a single large learner (e.g., a single deep learning model). To extend our approach for such models, it is necessary to introduce an additional mechanism that outputs tentative scores from intermediate layers. A critical challenge is mitigating the memory and time overhead introduced by this mechanism. Furthermore, a novel optimization algorithm tailored to this framework may be required, and promising directions include leveraging advanced generic optimization techniques, such as Bayesian optimization or Adam.

Additionally, our current optimization method does not always select the optimal intermediate layer thresholds, i.e., $\boldsymbol{\theta}$. While our experimental results demonstrate that selecting the best-performing thresholds from a set of candidates is sufficient to outperform existing LBFs in terms of memory efficiency and reject time, further improvements in threshold optimization could lead to even greater performance gains. Furthermore, while we currently discretize the false positive rates of Trunk Bloom filters, i.e., $\boldsymbol{f}^{(t)}$, at a certain granularity, future work could achieve better results by exploring more precise solutions.

Moreover, the configuration time of our approach is longer compared to existing methods such as sandwiched LBF or PLBF, which may pose a problem for specific applications. Therefore, accelerating the optimization process is another important direction for future work.

## 6    CONCLUSION

In this research, we proposed CLBF, solving two critical issues existing LBFs face. (1) By training a large machine learning model and reducing it optimally, CLBF achieves an optimal balance between model and filter sizes. (2) By branching based on tentative scores and the insertion of intermediate Bloom filters, CLBF significantly reduces reject time. As a result, CLBF not only broadens the applicability of LBFs but also establishes a strong foundation for addressing these issues.

REPRODUCIBILITY STATEMENT

We provide a comprehensive description of our method, with a detailed algorithm presented in Appendix A. The full source code, including dataset downloading, data preprocessing, model implementation, training, and evaluation, is available as supplementary material. Instructions for running the code are included in the `README.md` file within the package, ensuring that our results can be easily replicated.

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

---

**Algorithm 1** Key Insertion into CLBF

---

1: **Input:** Key $q$
2: **Function:** $\text{GetFBFIndex}(s)$ returns the index of FBF corresponding to the score $s$.

3: **for** $d = 1, 2, \ldots, D$ **do**
4:     $\text{TBF}_d.\text{Insert}(q)$
5:     $s \leftarrow \text{ML}_d(q)$
6:     **if** $d = D$ **then**
7:         $k \leftarrow \text{GetFBFIndex}(s)$
8:         $\text{FBF}_k.\text{Insert}(q)$
9:         **break**
10:    **if** $s \geq \theta_d$ **then**
11:       $\text{BBF}_d.\text{Insert}(q)$
12:       **break**

---

**Algorithm 2** Query Processing in CLBF

---

1: **Input:** Query $q$
2: **Output:** NotFound or Found
3: **Function:** $\text{GetFBFIndex}(s)$ returns the index of FBF corresponding to the score $s$.

4: **for** $d = 1, 2, \ldots, D$ **do**
5:     **if** $\text{TBF}_d(q) = \text{NotFound}$ **then**
6:         **return** NotFound
7:     $s \leftarrow \text{ML}_d(q)$
8:     **if** $d = D$ **then**
9:         $k \leftarrow \text{GetFBFIndex}(s)$
10:       **return** $\text{FBF}_k(q)$
11:    **if** $s \geq \theta_d$ **then**
12:       **return** $\text{BBF}_d(q)$

---

## A   ALGORITHM DETAILS

The pseudocode for the algorithm that inserts keys into the CLBF is shown in Algorithm 1. First, the key $q$ is inserted into the first Trunk Bloom filter, i.e., $\text{TBF}_1$. Next, the output score from the first machine learning model, i.e., $\text{ML}_1$, is obtained for the key $q$. If this score exceeds the threshold $\theta_1$ corresponding to $\text{ML}_1$, the algorithm branches to a Branch Bloom filter; $q$ is inserted into $\text{BBF}_1$, and the process terminates. Otherwise, $q$ is passed to the next depth. If $q$ does not branch into any Branch Bloom filters by the final depth $d = D$, it is inserted into the appropriate Final Bloom filter based on the final score. This process is repeated for all keys $q$ contained in $\mathcal{S}$, the set stored by the CLBF.

Next, the pseudocode for the query algorithm in the CLBF is shown in Algorithm 2. Similar to key insertion, the query $q$ is first checked against $\text{TBF}_1$. If $\text{TBF}_1$ returns a NotFound result, it is certain that $q \notin \mathcal{S}$, and this result is returned immediately. Otherwise, the output score from $\text{ML}_1$ is obtained for $q$. If this score exceeds the threshold $\theta_1$, the algorithm branches to the Branch Bloom filter; the algorithm queries $\text{BBF}_1$ for $q$, and the result from this filter is used as the final result. Otherwise, $q$ is passed to the next depth $d$. If $q$ does not branch into any Branch Bloom filters by the final depth $d = D$, it is queried against the appropriate Final Bloom filter based on the final score. This approach leverages the tentative and final scores of the machine learning models to provide fast query responses while preserving the false-negative-free property.

## B   ANALYSIS OF MODEL-FILTER MEMORY SIZE BALANCE

Here, we give observations of the model-filter memory size balance selected as a result of optimization by CLBF. Figure 7 illustrates the model-filter memory size balance selected by CLBF for each dataset and false positive rate configuration (here, we always set $\bar{D} = 100$). These results show that the optimal machine learning model size varies depending on the dataset and the target false positive

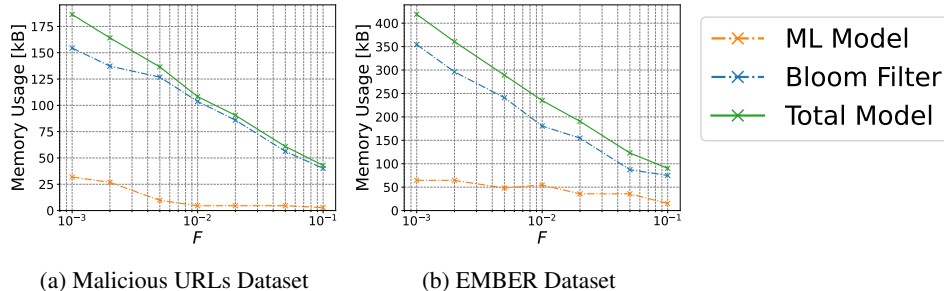

(a) Malicious URLs Dataset       (b) EMBER Dataset

Figure 7: The model-filter memory size balance selected by CLBF for various $F$ values: The optimal model size changes depending on the target false positive rate and the dataset.

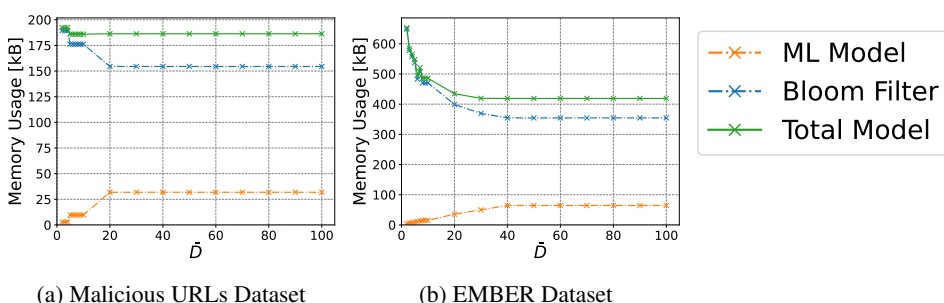

(a) Malicious URLs Dataset       (b) EMBER Dataset

Figure 8: The model-filter memory size balance selected by CLBF for various $\bar{D}$ values: Beyond a certain point, increasing $\bar{D}$ further does not change the results.

rate. For the Malicious URLs dataset, where learning is relatively easy, the proportion of memory occupied by the machine learning model is quite small, with the majority of the memory allocated to the Bloom filter. In contrast, for the EMBER dataset, which is relatively difficult to learn, the machine learning model occupies a relatively larger portion of the memory. Furthermore, across both datasets, as the target false positive rate decreases, the optimal machine learning model size tends to increase.

Next, we show in Figure 8 the relationship between the number of machine learning models used to construct CLBF, i.e., $\bar{D}$, and the model-filter memory size balance chosen as a result of optimization (we always set $F = 0.001$ here). When $\bar{D}$ is small, increasing $\bar{D}$ monotonically decreases the overall memory usage and monotonically increases the size of the machine learning model used. On the other hand, for a certain range of large values of $\bar{D}$, the size of the Bloom filter and the machine learning models used do not change. This is because CLBF only uses $D$, i.e., the number of machine learning models selected by optimization, from the given $\bar{D}$ machine learning models. Therefore, when using CLBF, it is assumed that users can obtain the optimal $D$ by making $\bar{D}$ larger than the expected optimal $D$.

In contrast, in the case of PLBF (Vaidya et al., 2021), setting a larger $D$ can lead to a decrease in memory efficiency. The relationship between the number of machine learning models given when constructing PLBF, i.e., $D$, and the memory usage of PLBF and its breakdown is shown in Figure 1 (we always set $F = 0.001$ here). As $D$ increases, of course, the memory usage of the PLBF machine learning models increases, because PLBF uses all of the $D$ machine learning models given. On the other hand, the memory usage of the backup Bloom Filter used by PLBF tends to decrease. This is because the accuracy of the machine learning model tends to improve as the size of the machine learning model increases, and even a small Bloom Filter can achieve the target false positive rate. The overall memory usage decreases in the range of a certain small $D$, and increases in the range of a certain large $D$. The optimal $D$ varies depending on the dataset, but in all cases, CLBF (shown as a red dot) selects a $D$ close to the optimal value.

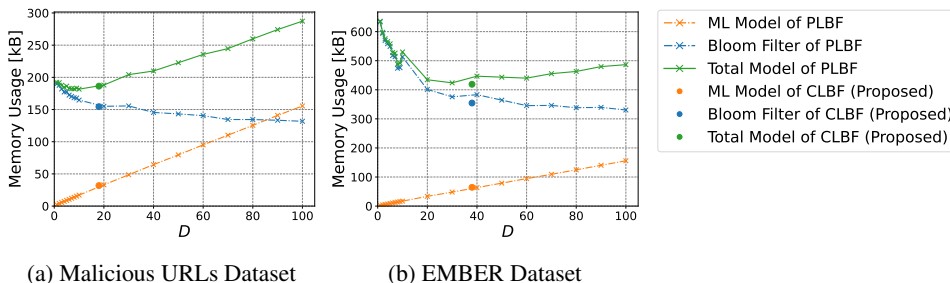

(a) Malicious URLs Dataset    (b) EMBER Dataset

Figure 9: The model and filter memory size achieved by PLBF for various $D$ values: Up to a certain point, increasing $D$ reduces the overall memory usage, but beyond that point, increasing $D$ starts to increase the overall memory usage.

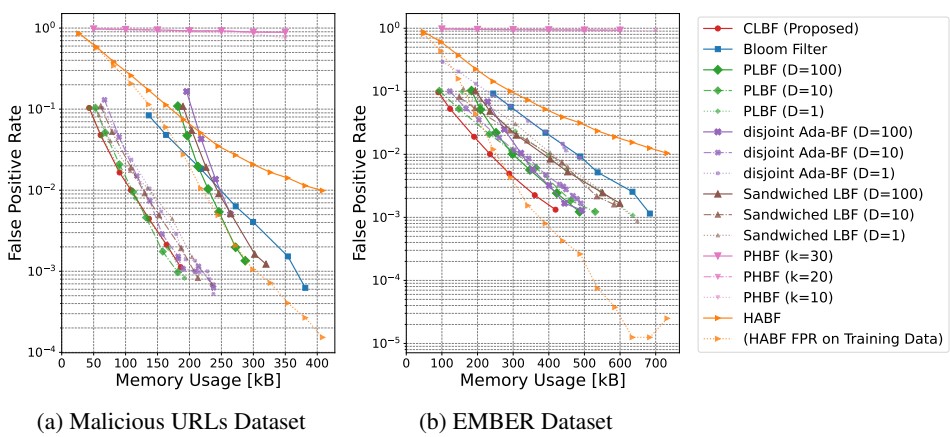

(a) Malicious URLs Dataset    (b) EMBER Dataset

Figure 10: Trade-off between memory usage and accuracy (lower-left is better).

## C COMPARISON EXPERIMENTS WITH OTHER BASELINES

Here, we present the comparison experiments with other baselines, which we omitted from the main text to avoid overly complicated result figures and to clearly demonstrate the effectiveness of our proposed method. In addition to the PLBF (Vaidya et al., 2021), the Sandwiched LBF (Mitzenmacher, 2018), and the Bloom filter (Bloom, 1970), we compare our CLBF with disjoint Ada-BF (Dai & Shrivastava, 2020) and Projection Hash Bloom Filter (PHBF) (Bhattacharya et al., 2022a), and Hash Adaptive Bloom Filter (HABF) (Xie et al., 2021b). We implemented the disjoint Ada-BF in C++ based on the Python implementation published by the authors (Dai & Shrivastava, 2024). For PHBF, we conducted experiments using the Python implementation published by the authors (Bhattacharya et al., 2022b), so note that the speed comparison for PHBF is **not** fair. The hyperparameter of PHBF, the sampling factor $s$, is always set to 10. For HABF, we used the C++ implementation published by the authors (Xie et al., 2021a).

### C.1 MEMORY AND ACCURACY

First, the trade-off between memory usage and accuracy for each method is shown in Figure 10. For CLBF, we always set $\bar{D} = 100$, and for PLBF, disjoint Ada-BF, and sandwiched LBF, we show the results for $D = 1, 10, 100$. For PHBF, the results for $k = 10, 20, 30$ are shown, where $k$ is the number of hash functions used in PHBF. For HABF, the results for `bits_per_key` $= 1, 2, \dots, 15$ are shown. For HABF, the false positive rate in the training data is also displayed, not just the false positive rate in the test data.

We can see that sandwiched LBF and disjoint Ada-BF always show inferior trade-offs compared to PLBF and CLBF with the same machine learning model size. In addition, the false positive rate for PHBF was always almost 1. This is thought to be because the mechanism of using the projection as

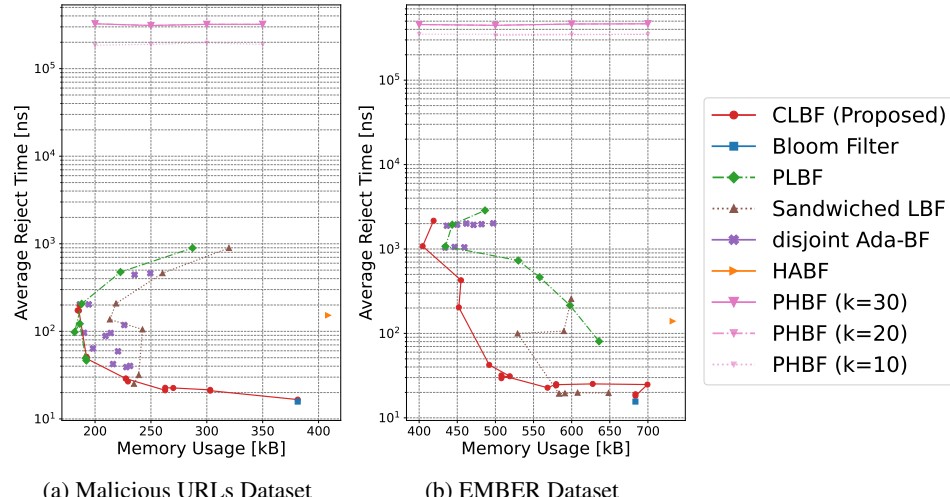

(a) Malicious URLs Dataset     (b) EMBER Dataset

Figure 11: Trade-off between memory usage and average reject time (lower-left is better).

a hash function results in false positives when there are keys in the set that have features similar to the non-key query. It is possible to improve the accuracy by increasing the PHBF hyperparameters $k$, i.e., the number of hashes, and $s$, i.e., the sampling factor, but this will lead to an increase in construction time. As we will show later, the construction time for PHBF is currently very long, so this approach is considered unacceptable.

## C.2 MEMORY AND REJECT TIME

Next, the trade-off between memory usage and reject time for each method is shown in Figure 11. For PLBF, sandwiched LBF, and Bloom filter, we show the results for $F = 0.001$. Because the hyperparameter that controls the accuracy of disjoint Ada-BF is the total memory usage (instead of the target false positive rate, as in PLBF and CLBF), it is difficult to compare them under consistent conditions. Therefore, for disjoint Ada-BF, we constructed models with various total memory usages and $D$s, and then plotted only those with a false positive rate close to $0.001$ (more precisely, greater than $0.75 \times 0.001$ and less than $1.25 \times 0.001$). For PHBF, we were unable to obtain a case with a sufficiently small false positive rate, so we have displayed the results for various memory usage when constructed with $k = 10, 20, 30$ by connecting the results for each $k$. Please note that it is **not** a fair comparison because the false positive rate is completely different between PHBF and other methods and because PHBF is implemented in Python while the others are implemented in C++.

For disjoint Ada-BF, when comparing with the same amount of memory usage, it was found that the reject time tended to be longer than PLBF. PHBF showed a very long reject time compared to the other methods. While the reject time for a Bloom filter was around $20\,\mathrm{ns}$ and LBFs were between $20\,\mathrm{ns}$ and $1,000\,\mathrm{ns}$, PHBF had a reject time of over $100,000\,\mathrm{ns}$.

## C.3 CONSTRUCTION TIME

Finally, the construction times for each method are shown in Figure 12. For PLBF, sandwiched LBF, and Bloom filter, we show the results for $F = 0.001$. For disjoint Ada-BF, we show the results for total memory usage is $1.6\,\mathrm{Mbit}$, and for HABF, we show for `bits_per_key` $= 8$. For PHBF, we show the results for $k = 10, 20, 30$.

We can see that the construction time for disjoint Ada-BF is almost the same as that for sandwiched LBF. This is because CLBF and PLBF use dynamic programming to find the optimal parameters, whereas disjoint Ada-BF uses heuristics to determine the parameters, as in sandwiched LBF. Also, we can see that the construction time for PHBF is much longer than those of the other methods. We can see that the construction time of HABF is much faster than that of the other methods. The construction time of HABF is $0.01$ to $0.08$ times that of PHBF.

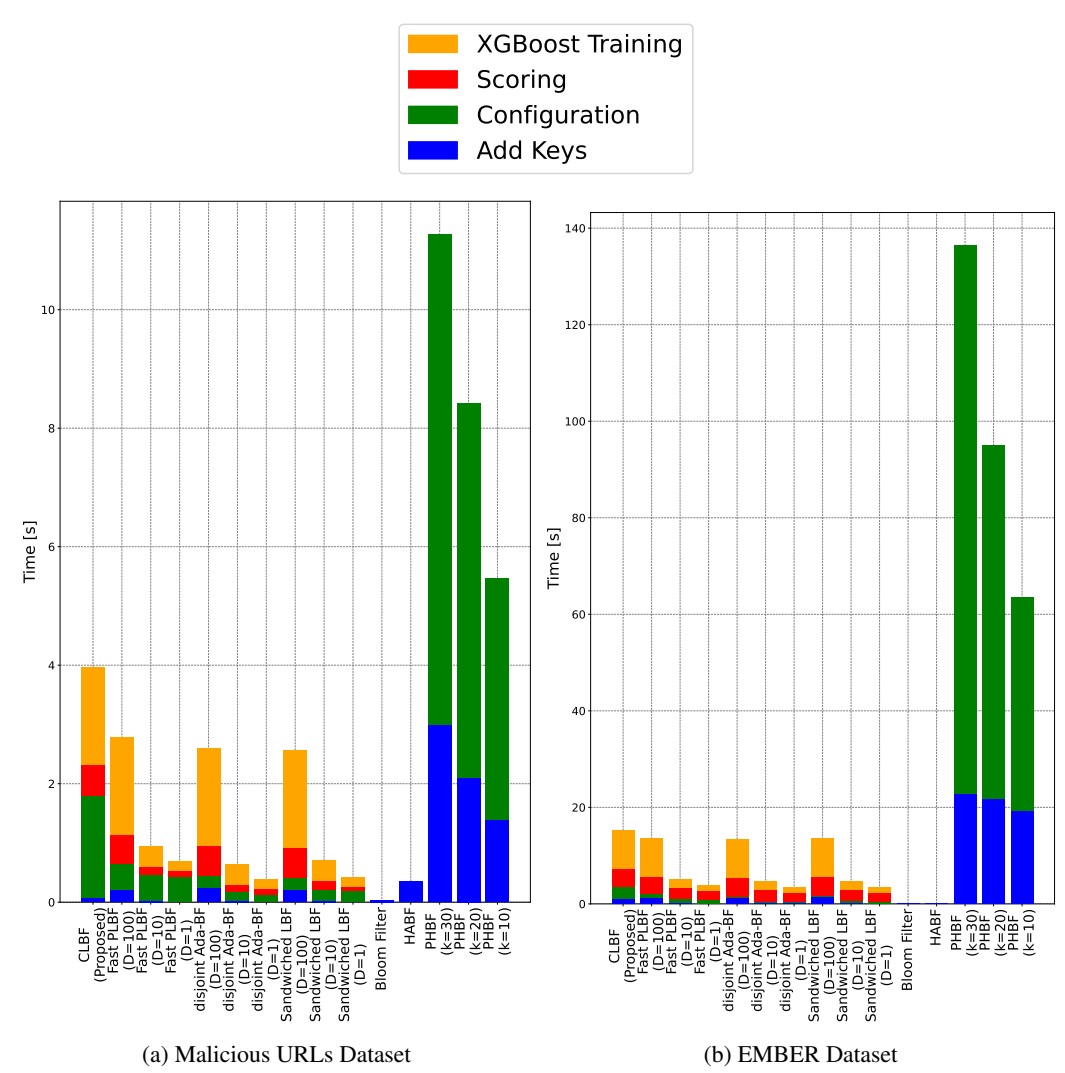

(a) Malicious URLs Dataset  (b) EMBER Dataset

Figure 12: Construction time.

# D  ABLATION STUDY ON HYPERPARAMETERS OF MACHINE LEARNING MODELS

Our CLBF improves memory efficiency and reduces reject time by optimizing the configuration, but it is possible to achieve the same effect by tuning the machine learning model (e.g., hyperparameter tuning, distillation, and quantization). These two approaches can be used at the same time. In other words, it is possible to construct a CLBF with superior performance by the optimization hyperparameters of the machine learning model followed by the optimization of the cascade structure. Here, we show the results of our observations on how the hyperparameters of the machine learning model XGBoost, in particular, the value of `max_depth`, affect the performance of LBF. The smaller the `max_depth`, the smaller each weak learner becomes, and while the size is smaller and inference time is shorter, the discriminative power of each weak learner becomes weaker.

The trade-off between memory usage and false positive rate for `max_depth` $= 1, 2, 4, 6$ is shown in Figure 13, and the trade-off between memory usage and reject time is shown in Figure 14, and the construction time is shown in Figure 15. We confirmed that the properties of CLBF, as shown in the main text, consistently appear for any value of `max_depth`: (1) better memory-accuracy trade-off than existing LBFs, (2) better memory-reject time trade-off than existing LBFs, (3) slightly (up to 1.8 times) longer construction time than existing LBFs.

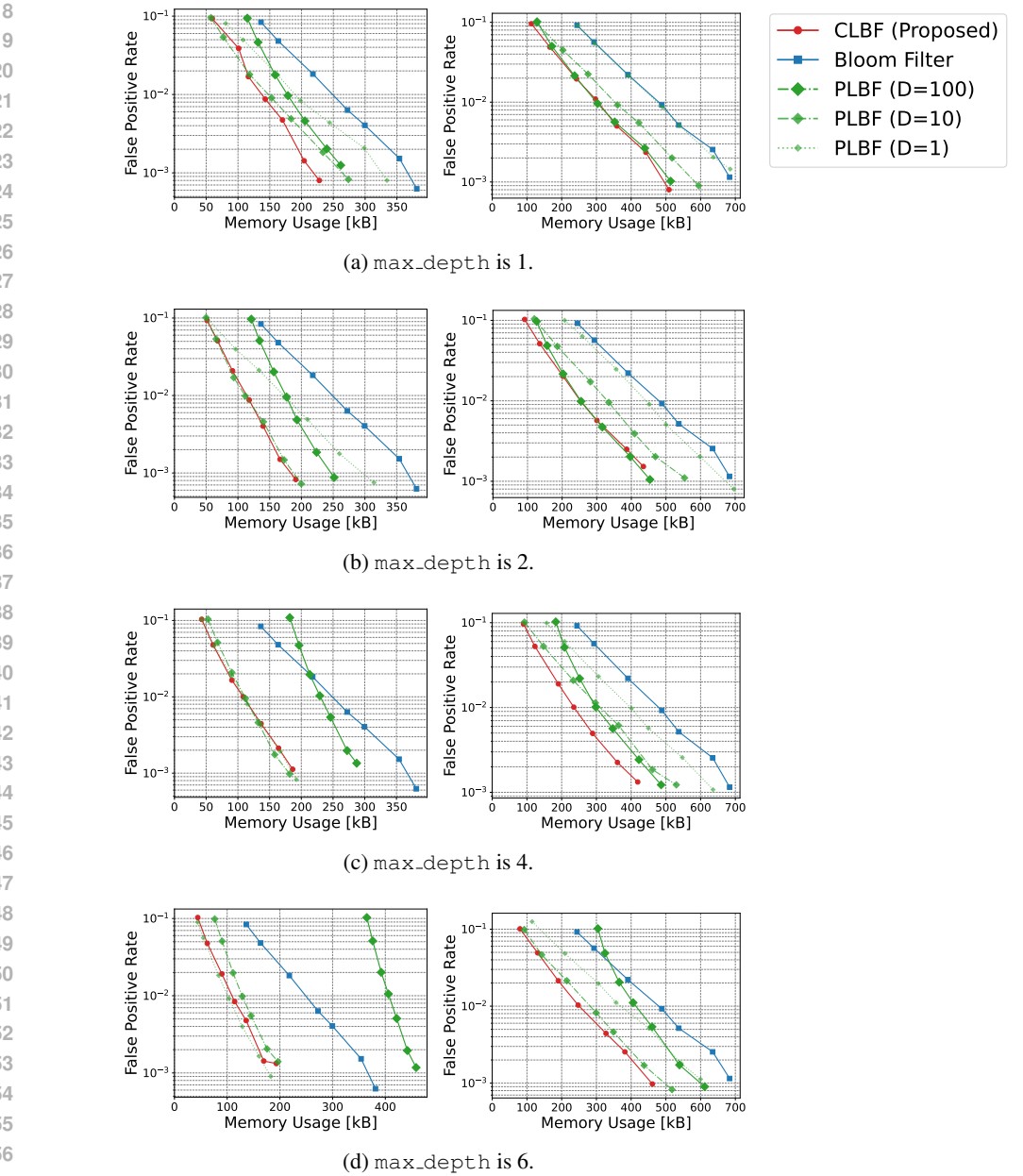

(a) max_depth is 1.

(b) max_depth is 2.

(c) max_depth is 4.

(d) max_depth is 6.

Figure 13: Trade-off between memory usage and accuracy (lower-left is better).

We find that there is a performance range that cannot be achieved by either the CLBF method or hyperparameter tuning alone, i.e., there is a performance that can only be achieved by using CLBF after selecting the ML model's hyperparameters appropriately. For example, in Figure 14, the performance of $(\text{Memory Usage}, \text{Reject Time}) = (500\,\text{kB}, 30\,\text{ns})$ can be achieved when $\text{max\_depth} = 4$ and CLBF is used, but it cannot be achieved when $\text{max\_depth} = 1$ or PLBF is used. These experimental results suggest the effectiveness of a hybrid method that combines machine learning model tuning and CLBF optimization.

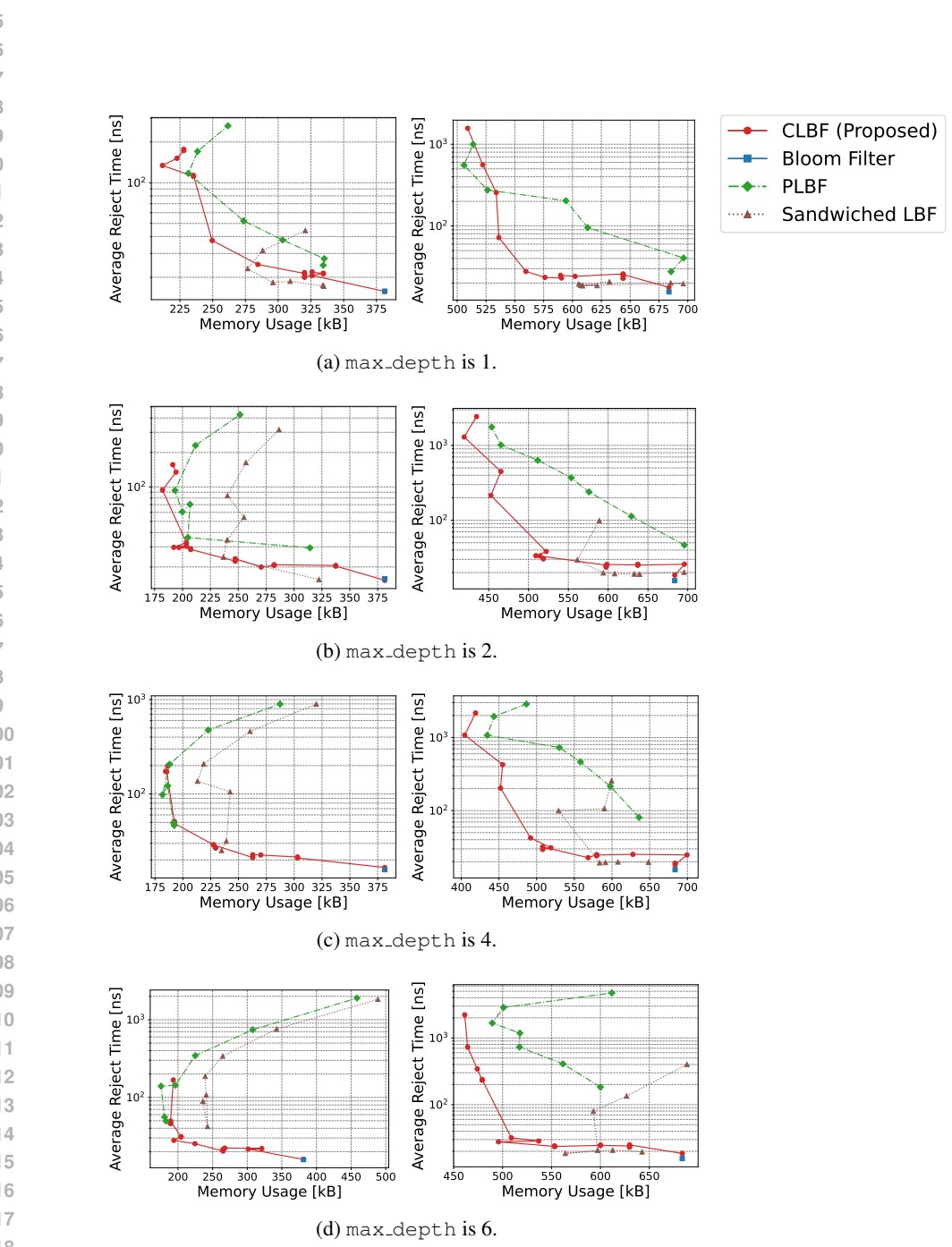

Figure 14: Trade-off between memory usage and average reject time (lower-left is better).

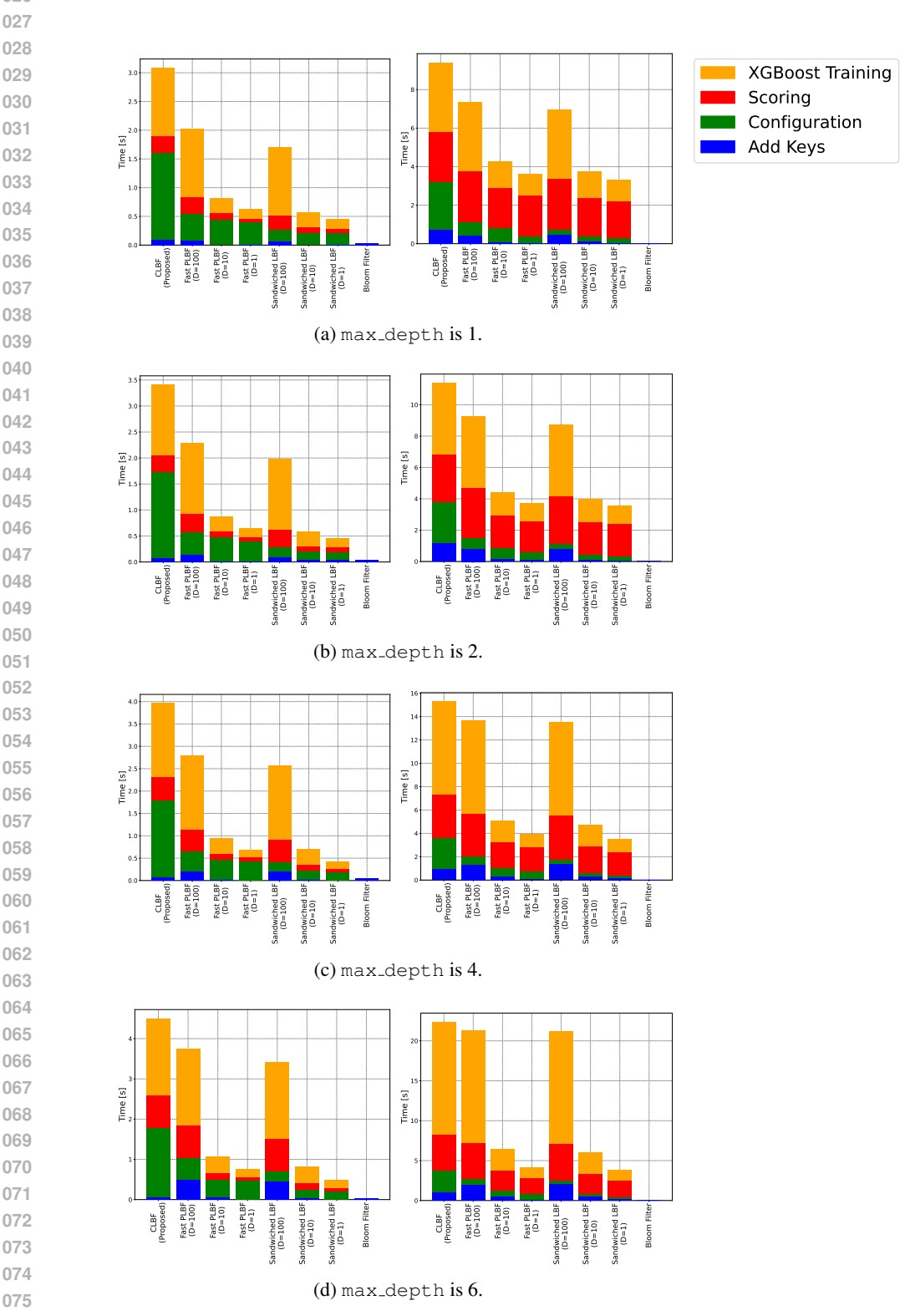

(a) max_depth is 1.

(b) max_depth is 2.

(c) max_depth is 4.

(d) max_depth is 6.

Figure 15: Construction time.

