# OpenReview forum: "Cascaded Learned Bloom filter for Optimal Model-Filter Size Balance and Fast Rejection"
_ICLR.cc/2025/Conference — Submitted to ICLR 2025_

### Official Review · Reviewer_U2zz · 2024-10-18

**Soundness:** 3
**Presentation:** 1
**Contribution:** 1
**Rating:** 1
**Confidence:** 5

**Summary:**

In this paper the authors present a new method for the set membership query that combines Bloom filters with machine learning models. Unlike the earlier definitions of learned Bloom filters (LBF) that typically combine a single learned model with a few Bloom filters, this paper uses an alternating sequence of learned models and Bloom filters to answer a query. The stated goals that the authors seek to achieve are the reduction of the memory footprint of the LBF and also reduction of the time to rejection of a negative query.

**Strengths:**

S1. The key strength of the paper is that the ML models are assumed to be weak learners. Since the use case of Bloom filters is generally in distributed databases and filesystems, often there are no strong patterns in the keys and so it is difficult to expect high accuracy (this conclusion is based on some work that has been done in my own group over the years). This comment highlights the significance of this work under review.

**Weaknesses:**

W1. Perhaps the most important weakness here is that the introduction of a multi-layered structure with great coding complexity and many learning models adds so much complexity to the structure that there is little to no chance that such a proposal will ever find its way into an actual file system. A quick look at Fig 1 and Fig 2 that have been placed side to side will show why.

W2. The tremendous increase in complexity of the solution is reflected in the construction time jump (see Fig 6). The authors need to ask themselves if such a massive increase over simple Bloom filters is justifiable. In the aftermath of Kraska's paper we seem to have forgotten that Bloom's idea was a lightweight and quick data structure to speed up set membership. In distributed filesystems there is a need to periodically rebuild all the set membership structures, once too many writes have been logged. I don't think the kind of construction times being shown here are acceptable.

W3. Important baselines are missing. The authors have not compared with the following methods that are competitors to Vaidya's PLBF:
- Zhenwei Dai and Anshumali Shrivastava. Adaptive Learned Bloom Filter
(Ada-BF): Efficient Utilization of the Classifier with Application to Real-Time
Information Filtering on the Web, NeurIPS 2020
- Arindam Bhattacharya, Chathur Gudesa, Amitabha Bagchi, Srikanta Bedathur.
New wine in an old bottle: Data-aware Hash Functions for Bloom Filters, VLDB 2022.

Both these papers measure memory used. While it can be argued that Vaidya et. al. have compared against Ada-BF already and beaten it on memory (but not comprehensively), the latter paper shows improved memory utilization over both Vaidya and Ada-BF. So there is no reason why it should not have been used as a baseline. The authors have cited the first paper but not the second.

W4. Memory-learning tradeoff not explored adequately. The authors mention that there has to be a tradeoff between memory and learning in terms of model size, but there is no experiment that investigates this tradeoff.

W5. Mathematica quantities not properly defined. For example, is Reject Time defined in terms of clock time?

**Questions:**

Q1. How would your method perform in a dynamic case where the set membership structures need to be reconstructed periodically?

Q2. How did you decide how much memory it to be assigned to Bloom filters and how much is to be kept for ML models?

---

> ### Author Response · Authors · 2024-11-23
>
> We sincerely appreciate the reviewer’s valuable comments and are glad to address their concerns.
>
> > Weakness 1: Perhaps the most important weakness here is that the introduction of a multi-layered structure with great coding complexity and many learning models adds so much complexity to the structure that there is little to no chance that such a proposal will ever find its way into an actual file system. A quick look at Fig 1 and Fig 2 that have been placed side to side will show why.
>
> We believe that our proposed method has the potential to be applied in contexts where simple data structures are required, such as file systems. Each ML model in our proposed method ($ML_1, ML_2, \dots ML_D$ in Fig. 2) can be very simple. In fact, the weak learner used as each ML model in this experiment was XGboost, which is essentially just a few conditional branches. Furthermore, it is also possible to use a simpler structure (e.g., just one conditional branch or just one arithmetic operation) as a "ML model". Even if each of these "ML models" is very small, CLBF selects the optimal number of "ML models" by considering reject time and memory efficiency, so the number of layers will not increase inappropriately. Therefore, users do not need to adjust the number of "ML models" themselves, and it is sufficient to prepare a relatively large number of "ML models" and pass them to CLBF. From this perspective, we believe that CLBF has the potential to be applied to contexts that require a simple structure.
>
>
> > Weakness 2: The tremendous increase in complexity of the solution is reflected in the construction time jump (see Fig 6). The authors need to ask themselves if such a massive increase over simple Bloom filters is justifiable. In the aftermath of Kraska's paper we seem to have forgotten that Bloom's idea was a lightweight and quick data structure to speed up set membership. In distributed filesystems there is a need to periodically rebuild all the set membership structures, once too many writes have been logged. I don't think the kind of construction times being shown here are acceptable.
>
> > Question 1: How would your method perform in a dynamic case where the set membership structures need to be reconstructed periodically?
>
> The frequency of reconstruction depends on the write frequency and the purpose of use, and we believe there are scenarios where LBFs (including CLBF) can be used effectively. For example, consider a Bloom filter constructed on a SSTable in a distributed database. If one SSTable holds 1,000,000 pieces of data and 1,000 pieces of data are written per second, the Bloom filter will be built every 1,000 seconds. If the construction time for LBF is around 4 to 15 seconds, as observed in our experiment, this construction time is considered to be almost negligible. Also, the construction time can be adjusted by changing the amount of data used for training and validation. On the other hand, the superior accuracy and memory efficiency of LBFs are valuable when filtering the SSTables to read. In this way, depending on the application, the longer construction time of LBF (including CLBF) compared to a simple Bloom filter can be considered acceptable.
>
> To emphasize this point, we have revised Section 4.3. The revised parts are highlighted in blue.

---

> > ### Comment · Reviewer_U2zz · 2024-11-23
> >
> > I'm sorry but this is not convincing. Please see the comparison between CLBF and just vanilla BF in your Fig 4 and Fig 6. For less than double memory commitment I can get construction time which appears to be 2 orders magnitude smaller. If I were a system designer, I would blindly choose vanilla BF in such a scenario. And this is not just to do with CLBF, all other learned BFs have the same problem.

---

> ### Author Response · Authors · 2024-11-23
>
> > Weakness 3: Important baselines are missing.
>
> First, we have omitted the evaluation of Ada-BF for the following two reasons. Firstly, Ada-BF always has worse memory efficiency than PLBF because Ada-BF is essentially the same architecture as PLBF, and while PLBF sets optimal parameters (i.e., thresholds and false positive rate of each backup Bloom filters) by solving optimization problems, Ada-BF uses heuristics and consequently set suboptimal parameters. Second, the reject time of Ada-BF is almost the same as PLBF as long as the same ML model is used. For these reasons, we omitted the evaluation of Ada-BF in our experiments. We conducted experiments with disjoint Ada-BF, which was proposed at the same time as Ada-BF and is known to have similar performance, and confirmed it (Appendix C).
>
> Second, although we were focusing on the Learned Bloom Filter that uses a ML model for binary classification, as you pointed out, we recognized the importance of positioning our research in a broader context that includes various types of Bloom Filters, including the Projection Hash Bloom Filter (PHBF) proposed by Arindam Bhattacharya et al. We have included PHBF to the Related Work section (Section 2), and also added a comparative experiment (Appendix C). We have also added Hash Adaptive Bloom Filter (HABF) to the Related Work Section and the comparative experiments in Appendix C. HABF is a method similar to PHBF in that it improves performance by optimizing the hash function using key and non-key information.
>
> The comparison experiments showed that our CLBF outperformed existing LBFs such as disjoint Ada-BF, PHBF, and HABF, and confirmed that the effectiveness of our method is maintained:
>
> - Trade-off between memory usage and accuracy (Fig. 10):
>   - Disjoint Ada-BF has worse memory efficiency than PLBF and CLBF.
>   - PHBF has a false positive rate of almost 1, no matter how large the memory usage was.
>   - HABF has a worse false positive rate than a simple Bloom filter. However, the HABF false positive rate for non-keys in the "training data", i.e. the data used for optimization, is much better than the false positive rate for "test data". In other words, HABF was found to be prone to overfitting.
> - Reject time (Fig. 11):
>   - Disjoint Ada-BF has a slightly longer reject time than CLBF (which has the same memory size).
>   - PHBF rejection time is about 100 times longer than other LBFs. (Note that while other methods are implemented in C++, PHBF uses the Python implementation published by the authors.)
>   - HABF has a longer reject time than other LBFs (which has the smaller memory size).
> - Construction time (Fig. 12):
>   - DisjointAda-BF required a slightly shorter construction time than Fast PLBF. Disjoint Ada-BF has almost the same construction time as Sandwiched LBF.
>   - PHBF required from 1.3x to 9.0x construction time as CLBF. (Note that while other methods are implemented in C++, PHBF uses the Python implementation published by the authors.)
>   - HABF’s construction time is 0.01 to 0.08 times that of PHBF.
>
> In conclusion, disjoint Ada-BF is inferior to CLBF in terms of memory efficiency and reject time, and is equivalent to Sandwich LBF in terms of construction time. Because the accuracy of PHBF is always bad, it was not possible to make a fair comparison. Although HABF has a very short construction time, its performance (in terms of accuracy for test data and reject time) was considerably worse than CLBF.

---

> > ### Comment · Reviewer_U2zz · 2024-11-23
> >
> > Thank you for adding these baselines. Your new experiments have made the picture more complete.

---

> ### Author Response · Authors · 2024-11-23
>
> > Weakness 4: Memory-learning tradeoff not explored adequately. The authors mention that there has to be a tradeoff between memory and learning in terms of model size, but there is no experiment that investigates this tradeoff.
>
> Classifier-based LBF consists of one binary classifier and one or more Bloom filters, and in our paper we state that there is a trade-off between the memory size of the ML model used in LBF and the memory size of the Bloom filters used in LBF (39L-41L in the main text). This is supported both theoretically and experimentally.
>
> First, the theoretical support is provided by the PLBF paper. It is known that the better the model accuracy (in terms of Kullback–Leibler divergence), the smaller the Bloom filter size required to achieve the target false positive rate. Therefore, under the natural assumption that "the larger the model size, the better the model accuracy," we can deduce that there is a trade-off between bloom filter memory size and ML model size.
>
> In addition, our experiments also show that this trade-off exists (Figure 4). If there were no trade-off between Bloom filter memory size and ml model size, then the larger or smaller the ml model size, the smaller the overall memory usage should be. However, this is not the case, so we can see that there is a trade-off between Bloom filter memory size and ml model size.
>
> Furthermore, we have included the results of additional experiments in Appendix B, in which we observed the bloom filter memory size and ML model size when the $D$ of PLBF was changed in various ways. From the appearance of Figure 9, it can be seen that there is a trade-off between the bloom filter memory size and ML model size.
>
>
> > Weakness 5: Mathematica quantities not properly defined. For example, is Reject Time defined in terms of clock time?
>
> Yes, Reject Time refers to the actual elapsed clock time measured from when the data structure receives a query to when it returns an answer of rejection. Additionally, the time taken by each ML model to make inferences, denoted as $\mathrm{Time}(ML_i)$, is also measured as the actual clock time required for the model to perform inferences in this experiment.
>
>
> > Question 2: How did you decide how much memory it to be assigned to Bloom filters and how much is to be kept for ML models?
>
> CLBF automatically determines the optimal memory allocation between the Bloom filter and the ML model using dynamic programming, in order to achieve good accuracy and short reject time. Dynamic programming is performed to minimize the value defined in Equation (1), where the $(\mathrm{Memory~Size})$ term represents the total memory allocated to both the Bloom filters and the ML model, as defined in Equation (2). Unlike existing LBFs, which aim to minimize the memory allocated to the Bloom filters while fixing the ML model, our approach optimizes the size of the ML model as a variable parameter.

---

> > ### Comment · Reviewer_U2zz · 2024-11-23
> >
> > Thanks for your responses. I'm sorry but I don't think that "the larger the model size, the better the accuracy" is a rigorous statement. In general the relationship between accuracy and model size is a complex matter that varies from model to model and from dataset of dataset even for the same model. I am not sure Figure 4 is the answer to my W4.

---

> ### Author Response · Authors · 2024-11-25
>
> We sincerely appreciate your response. We greatly appreciate the opportunity to address your concerns and provide further clarifications.
>
> > I'm sorry but I don't think that "the larger the model size, the better the accuracy" is a rigorous statement. In general the relationship between accuracy and model size is a complex matter that varies from model to model and from dataset of dataset even for the same model.
>
> In the part of our paper that mentions this tradeoff (Introduction: L39-L41), we claimed only the tendency, but in our earlier response, we used a simplified statement for clarity. We apologize for the lack of rigor in our wording.
>
> Our main point is not that there is a strict tradeoff between the size of an ML model and its accuracy. Rather, we want to emphasize that minizing the total memory usage of the LBF (the sum of the memory used by the ML model and the memory used by the Bloom filter within the LBF) is achieved by treating the size of the ML model as a variable to be optimized. We referred to this tradeoff to provide an intuitive explanation, but we realize that this could lead to the misconception that our method depends on this tradeoff. To clarify, we have edited (and highlighted in blue) a statement in the Introduction to refer to the complexity of the relationship between model size and accuracy.
>
> Our approach does not rely on the tradeoff between ML model size and accuracy. Instead, it determines the appropriate model size by analyzing the tentative outputs of the ML model on the validation data. Therefore, even if this assumption does not hold, our method can still select an appropriate model size.
>
> To further clarify that our method does not rely on this tradeoff, we provide the following mathematical explanation. Let us assume an ML model with size $m$ and accuracy $a$ (measured by Kullback-Leibler divergence). It is known that for a PLBF to achieve a false positive rate $f$, the size of the backup Bloom filters required can be approximated as:
>
> $$ c n (\log_2 (1/f) - a) $$
>
> where $c$ is a constant and $n$ is the size of the set maintained by the PLBF. Thus, the total memory size of the PLBF is given by:
>
> $$ m + c n (\log_2 (1/f) - a) $$
>
> Now, suppose we slightly increase the size of the ML model by $\Delta m > 0$, resulting in an accuracy increase of $\Delta a$ ($\Delta a$ may be either positive or negative). The corresponding change in the total memory size of PLBF is given by:
>
> $$ \Delta m -  c n \Delta a. $$
>
> To put it simply (omitting details such as reject time or branching), our algorithm essentially determines whether $\Delta m -  c n \Delta a > 0$. If this condition holds, it selects the smaller ML model; otherwise, it chooses the larger ML model. This ensures that even if the assumption "the larger the model size, the better the accuracy" does not hold (that is, if increasing the model size decreases accuracy, i.e., $\Delta a < 0$), our method selects smaller ML Model and avoids an inappropriately large ML model because $\Delta m -  c n \Delta a > 0$ holds when $\Delta a < 0$.
>
> > I am not sure Figure 4 is the answer to my W4.
>
> Indeed, Figure 4 does not directly show the tradeoff, so perhaps it should not have been mentioned here. Figure 9 shows this tradeoff more directly, so please refer to that (we have replaced Figure 9 to make it more easy to confirm the tradeoff). At least in this experiment, we can see that as the number of the weak learners increases, the accuracy tends to increase, and the size of the Bloom filter required by PLBF tends to decrease.
>
> > Thank you for adding these baselines. Your new experiments have made the picture more complete.
>
> Your constructive comments have greatly contributed to improving the quality and comprehensiveness of our work. We sincerely appreciate your detailed suggestions, which helped us address critical issues and enhance our study.

---

> ### Author Response · Authors · 2024-11-25
>
> > For less than double memory commitment I can get construction time which appears to be 2 orders magnitude smaller. If I were a system designer, I would blindly choose vanilla BF in such a scenario.
>
> We fully agree that the classical Bloom filter is important in cases where construction time and simplicity are emphasized. However, we believe that this fact is compatible with the fact that LBF is a valuable alternative when computation time is relatively less important.
>
> Instead of comparing memory usage under the same false positive rate, comparing false positive rates under the same memory constraints reveals the value of the LBF. In Figure 4, we observe that the CLBF achieves 10x–20x smaller false positive rates than the Bloom filter for the same memory usage. This significant improvement in accuracy should not be overlooked. That is, for systems with severely constrained memory, such as IoT devices, where the goal of the Bloom filter is to maximize accuracy under fixed memory budgets, the LBF presents a compelling option.
>
> The accuracy improvement of the (learned) Bloom filter provides substantial value in real-world scenarios where Bloom filters are commonly used. For example, as mentioned in our response to Reviewer YPfA’s "Weakness 2" and "Questions 2," consider a system where positive responses from the Bloom filter trigger costly subsequent processes (e.g., "exact" membership checks). Reducing the false positive rate by 90% leads directly to a 90% reduction in unnecessary downstream processing for queries not in the set. This can significantly enhance the overall system performance. Furthermore, in the case of IoT devices, for example, it is quite possible that the ML training and Bloom filter construction are completed in advance, and only the resulting binary is distributed (in fact, this is how ML models are often deployed on IoT devices). In such cases, LBF is not complex from the user's point of view, and the improvement of LBF accuracy is important.

---

### Official Review · Reviewer_YPfA · 2024-10-31

**Soundness:** 2
**Presentation:** 3
**Contribution:** 1
**Rating:** 3
**Confidence:** 4

**Summary:**

The paper presents the Cascaded Learned Bloom Filter (CLBF), a learned Bloom filter design aiming to optimize the balance between memory usage and reject time. The CLBF achieves these goals using dynamic programming and by designing a cascaded structure combining machine learning models and Bloom filters. Experimental evaluations on real-world datasets demonstrate CLBF’s improvements in reject time and memory efficiency compared to previous state-of-the-art methods.

**Strengths:**

1- The paper effectively summarizes and describes prior research on learned Bloom filters, providing a clear understanding of existing work and where CLBF differs. For example, CLBF introduces a unique combination of sandwiched BF and PLBF structures in a cascade, allowing a flexible balance between model and filter sizes and distinguishing it from prior approaches.

2- The experiments cover multiple datasets and baselines, which provides a well-rounded analysis of CLBF's effectiveness and limitations across diverse configurations.

**Weaknesses:**

1- The paper’s primary contribution seems minimal, as it largely combines established techniques from sandwiched Bloom filters and PLBF within a cascade structure. This approach suggests an incremental rather than a groundbreaking contribution.

2- The emphasis on optimizing reject time raises applicability concerns. There is a lack of discussion about real-world applications where reduced reject time is crucial, as well as cases where it might be less relevant. More critically, the cascade structure may increase query time for elements in the set, impacting overall efficiency. A fair comparison would benefit from including average query time metrics across different query types.

3- The dynamic programming approach, while effective, is computationally intensive. Any change in parameters would require re-optimization, and even without changes, the initial construction time is high (as illustrated in Figure 6). This may limit the scalability and practicality of CLBF in time-sensitive applications.

**Questions:**

1- Could the design be made general for any learning-augmented application, extending beyond Bloom filters?

2- What are real-world applications where reduced reject time is prioritized over faster query times for present keys? How could the cascade depth be controlled based on the target query time for these applications?

3- Given the variety of ML models with diverse performance profiles (e.g., different sizes or query times), how adaptable is the proposed design? Would the structure or configuration require major adjustments for different ML model types?

Minor typos:

(1) line 266 includes extra "

---

> ### Author Response · Authors · 2024-11-23
>
> We appreciate the reviewer's insightful comments and are happy to address their concerns.
>
> > Weakness 1: The paper’s primary contribution seems minimal, as it largely combines established techniques from sandwiched Bloom filters and PLBF within a cascade structure. This approach suggests an incremental rather than a groundbreaking contribution.
>
> It is true that the architecture of our proposed CLBF is a generalization of the existing LBF (Sandwiched LBF and PLBF) architecture, but the cascaded structure is completely novel technology. In addition, the dynamic programming method that optimizes each parameter under this generalized architecture should also be completely novel. We see the fact that CLBF is a generalization of existing LBF as an advantage of CLBF, because by selecting the optimal configuration under this generalized architecture, it is possible to surpass existing LBF and reach a performance that could not be achieved with existing LBF.
>
> Also, our problem setting (automatic tuning to optimize the filter-model size balance and minimize reject time) is novel, and as a result, the optimization method is also novel. Existing LBFs were proposed under the problem setting that arranging Bloom filters to achieve good memory efficiency when a fixed ML model is given. In contrast, our method differs from existing LBFs in at least two points: (1) it also optimizes the size of the ML model, and (2) it optimizes not only for memory efficiency but also for reject time. None of the existing LBF studies have proposed a method to adaptively determine the size of the ML model itself or to automatically optimize its configuration concerning rejection time. Therefore, we believe that the problem setting and solution of CLBF are very different from those of existing methods.

---

> ### Author Response · Authors · 2024-11-23
>
> > Weakness 2: The emphasis on optimizing reject time raises applicability concerns. There is a lack of discussion about real-world applications where reduced reject time is crucial, as well as cases where it might be less relevant. More critically, the cascade structure may increase query time for elements in the set, impacting overall efficiency. A fair comparison would benefit from including average query time metrics across different query types.
>
> > Questions 2: What are real-world applications where reduced reject time is prioritized over faster query times for present keys? How could the cascade depth be controlled based on the target query time for these applications?
>
> Because the Bloom filter's response includes false positives, when the Bloom filter responds that "the query $q$ is included in the set $\mathcal{S}$", there is often a subsequent process to confirm the correct answer using heavy calculations. On the other hand, because the Bloom filter's response of "$q \notin \mathcal{S}$" does not include any errors, there is no need for a confirmation process. Therefore, while the time taken for "$q \in \mathcal{S}$" response is unlikely to become a bottleneck in application, the time for "$q \notin \mathcal{S}$" response is often be a bottleneck.
>
> As an example, consider the case of using a Bloom filter for local filtering when a browser determines whether a URL is malicious. When a user attempts to access a URL $q$, the browser first queries the local Bloom filter constructed over the set of known malicious URLs $\mathcal{S}$. If the Bloom filter responds with $q \notin \mathcal{S}$, the browser can immediately conclude that $q$ is not in the known malicious URLs and allow the user to access the URL, because the Bloom filter guarantees no false negatives. On the other hand, if the Bloom filter responds with $q \in \mathcal{S}$, the browser must send the query to the server to verify the "exact" membership. In such cases, even if the Bloom filter takes slightly longer to process queries for present keys, it is not a significant issue since the verification step, which is much more time-consuming, is the bottleneck. However, the Bloom filter's reject time is critical, as it directly impacts the response time for queries corresponding to URLs not in the set.
>
> The above example can be explained using the following mathematical formula. Let $t_a$ and $t_r$ be the time that the Bloom filter takes to respond for key and non-key, respectively. Also, let $F$ be the false positive rate of the Bloom filter and $T$ be the time for the subsequent heavy process (satisfying $t_a \ll T$, $t_r \ll T$, and $F \sim 0$). Then,
> - The average time taken for a "exact" membership query when the query is in the set is $t_a + T$, and noting that $t_a \ll T$, this is almost equal to $T$.
> - The average time taken for a “strict” membership query when the query is not in the set is $(1 - F) t_r + F (t_r + T)$, and noting that $t_r \ll T$ and $F \sim 0$, this is almost equal to $t_r + FT$.
>
> From the above, $t_a$ is not a bottleneck (as long as $t_a \ll T$), but $t_r$ is a bottleneck when $t_r > FT$.
> We have revised Section 1 to make this point clearer. The revised parts are highlighted in blue.

---

> ### Author Response · Authors · 2024-11-23
>
> > Questions 1: Could the design be made general for any learning-augmented application, extending beyond Bloom filters?
>
> Thank you for your insightful comments. We believe that the problem CLBF addresses, i.e., optimizing the size of the ML model itself and avoiding heavy inference of the ML model, often appears in other learning-augmented applications. Also, we believe that the solution of CLBF, i.e., reducing a large ML model to an appropriate size and inserting branches using the tentative output of the ML model, can be applicable to various learning-augmented applications. However, in order to apply this idea to individual learning-augmented applications, we expect that it will be necessary to carefully consider each individual problem, as in the case of CLBF.
>
>
> > Weakness 3: The dynamic programming approach, while effective, is computationally intensive. Any change in parameters would require re-optimization, and even without changes, the initial construction time is high (as illustrated in Figure 6). This may limit the scalability and practicality of CLBF in time-sensitive applications.
>
> Certainly, the construction time for CLBF is much longer than that for Bloom filter, so it is not suitable for construction time-sensitive applications, but as we wrote in Section 4.3, we think that CLBF is often acceptable in scenarios where the construction time for existing LBF is acceptable. Bloom Filters are a data structure that is fast to construct, and can be constructed in around 20ms in our experiment. On the other hand, LBF takes 500ms to construct, i.e., around 25 times longer than Bloom filter, even in the case where the construction time is the shortest (using a very small ML model and Sandwiched LBF, a LBF with short construction time). From these observations, it can be thought that in many scenarios where LBF is used, it is unlikely that construction times as fast as those of Bloom Filters will be required.
>
> For example, in the case of using (Learned) Bloom Filters for local filtering of malicious URLs, it is generally acceptable for the construction process to take a significant amount of time. This is because the (Learned) Bloom Filter, once constructed, can be reused for an extended period, as the set of malicious URLs is not likely to change rapidly. Our CLBF requires a maximum of 1.41 times longer construction time than the Sandwiched LBF (compared in the situation of receiving the same ML model as the CLBF), so we believe that if the construction time of the Sandwiched LBF is acceptable, the construction time of the CLBF will also be acceptable in many cases. In other words, it is unlikely that the scalability and practicality of the CLBF will be narrower than that of the existing LBF due to the construction time.
>
> To emphasize this point, we have revised Section 4.3. The revised parts are highlighted in blue.
>
>
> > Question 3: Given the variety of ML models with diverse performance profiles (e.g., different sizes or query times), how adaptable is the proposed design? Would the structure or configuration require major adjustments for different ML model types?
>
> Even if the given $\bar{D}$ ML models that have different sizes/query times (i.e., $\mathrm{Size}(ML_d)$ and $\mathrm{Time}(ML_d)$ vary depending on $d$), our optimization method can be applied utilizing the information. This is expressed in equations (8) and (9), which represent the calculation method for dynamic programming, and the size and query time of each ML model are expressed in a form that depends on $d$, and it can be seen that even if $\mathrm{Size}(ML_d)$ and $\mathrm{Time}(ML_d)$ differ depending on $d$.

---

> > ### Comment · Reviewer_YPfA · 2024-12-02
> >
> > Thank you for your response. I appreciate the clarifications, but I remain concerned about the novelty of the approach. Additionally, the dynamic programming method seems computationally intensive, which may make the vanilla Bloom Filter a more practical choice in many scenarios where we care about the rejection time.

---

> ### Author Response · Authors · 2024-12-04
>
> We sincerely appreciate your response and are glad to address your concerns.
>
> > I remain concerned about the novelty of the approach.
>
> Our approach has novelty in both perspectives of problem setting and method:
> - Our problem setting consists of the following two parts, and neither of them has been tackled in existing research:
>   - Automatic tuning to optimize the filter-model size balance.
>   - Automatic tuning to minimize reject time.
> - Our proposed CLBF, whose architecture is an extension of the existing LBF architectures, has novelty in its cascade architecture and is also novel in its optimization method. Also, this novelty has produced the following performance improvements compared to existing LBF:
>   - Reduction in memory usage of up to 24%.
>   - Reduction in reject time of up to 14 times.
>
>
> > Additionally, the dynamic programming method seems computationally intensive, which may make the vanilla Bloom Filter a more practical choice in many scenarios where we care about the rejection time.
>
> Dynamic programming is an algorithm for construction, so the long calculation time of dynamic programming does not lead to a long reject time, i.e., the time to respond for a non-key query. Instead, this dynamic programming is performed to improve the reject time (and memory efficiency). CLBF shows performance that is comparable to or much better than Bloom filters in terms of reject time and memory efficiency:
> - Figure 4 shows that CLBF achieves 10 to 20 times higher precision than a Bloom filter with the same memory consumption.
> - From Figure 5, we can see that CLBF achieves a 25% to 40% reduction in memory consumption with a reject time about twice that of a Bloom filter.
>
> Therefore, our CLBF can be preferred over a Bloom filter in sufficiently broad scenarios where reject time and/or memory efficiency is prioritized.
>
> Also, the long construction time of CLBF is acceptable in scenarios where reconstructions are not required very often. Examples of such scenarios include:
> - The local filtering of malicious URLs that we mentioned in the previous response.
> - The Bloom filter on SSTable that we mentioned in our response to reviewer U2zz. Here, we quantitatively demonstrate that CLBF's long construction time poses few problems in this scenario, where some degree of frequent reconstruction is required, so we hope you will refer to it.

---

### Official Review · Reviewer_whcR · 2024-11-01

**Soundness:** 3
**Presentation:** 4
**Contribution:** 4
**Rating:** 6
**Confidence:** 4

**Summary:**

This article introduces a Cascaded Learned Bloom Filter (CLBF). CLBF combines the advantages of machine learning and traditional Bloom filters, aiming to address two critical issues in existing Learned Bloom Filters (LBF): the optimal balance between the size of the machine learning model and the Bloom filter, and the effective minimization of rejection time. CLBF employs a cascade structure with multiple machine learning models and multiple Bloom filters arranged alternately. The goal is to minimize memory usage and rejection time under the constraint that the false positive rate does not exceed F. CLBF uses dynamic programming optimization methods to automatically select configurations, achieving an optimal balance between model and filter sizes, and minimizing rejection time. Experiments show that CLBF reduces memory usage by 24% and rejection time by 14 times compared to the state-of-the-art Learned Bloom Filter (PLBF).

**Strengths:**

1. The article proposes a novel structure of CLBF that optimizes memory usage and query performance by cascading multiple machine learning models and Bloom filters. It utilizes dynamic programming to automatically select the optimal configuration, reducing the manual parameter tuning effort and potentially finding better solutions.
2. By adjusting hyperparameters, CLBF can trade off between memory efficiency and rejection time to adapt to different application scenarios. The article conducts extensive experiments with real-world datasets, validating the effectiveness of CLBF.
3. The paper provides a clear description of the CLBF's implementation, accompanied by a thorough theoretical analysis.

**Weaknesses:**

1. The article primarily focuses on the application of a single machine learning model, XGBoost, which limits the exploration of how different models might contribute to the performance of CLBF. A comparison with a variety of machine learning models could provide a more comprehensive understanding of CLBF's versatility and robustness across different algorithmic approaches.
2. While the article effectively demonstrates the benefits of CLBF, it would be beneficial to explore whether fine-tuning the hyperparameters of the machine learning model or alternative constructions of cascaded Bloom filters could yield similar or superior optimization results. This discussion could offer additional insights into the factors that contribute to the overall performance of CLBF.

**Questions:**

1. Is it possible that, instead of employing a cascaded structure, optimizing the hyperparameters of the machine learning model within a Learned Bloom Filter could also achieve similar performance enhancements?

2. It raises an interesting question whether embedding a multi-layer deep learning model within the initial machine learning framework could further enhance optimization. Could such a multi-layer structure, potentially through algorithms like Bayesian Optimization, or Adam, autonomously identify optimal network parameters to achieve enhanced model optimization?

3. Is it possible to use a combination of a machine learning model with hierarchical Bloom filters for optimization? For example, using one Bloom filter to filter out most elements, and then using a second, more precise Bloom filter for final verification.

---

> ### Author Response · Authors · 2024-11-23
>
> We are thankful for the reviewer’s constructive feedback and are happy to address their inquiries.
>
> > Weakness 1: The article primarily focuses on the application of a single machine learning model, XGBoost, which limits the exploration of how different models might contribute to the performance of CLBF. A comparison with a variety of machine learning models could provide a more comprehensive understanding of CLBF's versatility and robustness across different algorithmic approaches.
>
> Thank you for your insightful comments. Indeed, it is important to evaluate CLBF using various machine learning models. CLBF optimize its configuration using the information about the received machine learning models (the size $\mathrm{Size}(ML_i)$ and the inference time $\mathrm{Time}(ML_i)$), so we believe that CLBF it can select an appropriate architecture that leverages the characteristics of the machine learning models.
>
> To quickly confirm this, we constructed CLBF and existing LBFs using XGBoost with various hyperparameters (Appendix D). We conducted experiments by changing the XGBoost hyperparameter max_depth. The smaller the max_depth, the smaller each weak learner becomes, and while the size is smaller and inference time is shorter, the discriminative power of each weak learner becomes weaker. The results can be summarized as follows:
>
> - In terms of both memory efficiency and reject time, our CLBF consistently achieved performance that was equal to or better than existing LBFs, regardless of max_depth (Figures 13 and 14).
> - In terms of construction time, the construction time for CLBF was up to 1.8 times longer than that of existing LBFs (Figure 15).
>
> This suggests that our CLBF consistently outperforms existing LBFs regardless of the size, inference time, or discriminative power of the machine learning model it received.
>
> > Weakness 2: While the article effectively demonstrates the benefits of CLBF, it would be beneficial to explore whether fine-tuning the hyperparameters of the machine learning model or alternative constructions of cascaded Bloom filters could yield similar or superior optimization results. This discussion could offer additional insights into the factors that contribute to the overall performance of CLBF.
>
> > Question 1: Is it possible that, instead of employing a cascaded structure, optimizing the hyperparameters of the machine learning model within a Learned Bloom Filter could also achieve similar performance enhancements?
>
> It is possible that similar performance improvements (in terms of memory efficiency and reject time) can be achieved by changing the hyperparameters of the machine learning model. Such a method can be used in conjunction with our optimization method. In other words, it is possible to construct an CLBF with superior performance by the optimization hyperparameters of the machine learning model followed by the optimization of the cascade structure.
>
> The experimental results in Appendix D demonstrate the possibility of improving performance using such a hybrid method. In Appendix D, the experimental results of the performance evaluation of CLBF constructed using XGBoost with various max_depth values are shown. These results suggest that there is a performance range that cannot be achieved by either the CLBF method or hyperparameter tuning alone, i.e., there is a performance that can only be achieved by using CLBF after selecting the ML model's hyperparameters appropriately. For example, in Figure 14(a) and (c), the performance of (Memory Usage, Reject Time) = (500kB, 30ns) can be achieved when max_depth = 4 and CLBF is used, but it cannot be achieved when max_depth = 1 or PLBF is used. These experimental results suggest the effectiveness of a hybrid method that combines machine learning model tuning and CLBF optimization.
>
>
> > Question 2: It raises an interesting question whether embedding a multi-layer deep learning model within the initial machine learning framework could further enhance optimization. Could such a multi-layer structure, potentially through algorithms like Bayesian Optimization, or Adam, autonomously identify optimal network parameters to achieve enhanced model optimization?
>
> Thank you for your insightful question. Indeed, a novel optimization algorithm tailored to the framework may be required, and promising directions include leveraging advanced generic optimization techniques, such as Bayesian optimization or Adam. We have added the mention of this possible approach in the Limitations chapter (Section 5).

---

> ### Author Response · Authors · 2024-11-23
>
> > Question 3: Is it possible to use a combination of a machine learning model with hierarchical Bloom filters for optimization? For example, using one Bloom filter to filter out most elements, and then using a second, more precise Bloom filter for final verification.
>
> Yes, it is possible that the performance of LBF can be improved by using such hierarchical Bloom filters. From the perspective of the trade-off between memory usage and accuracy, it is known that there are no benefits to be gained from using only Bloom filters in a hierarchical manner. However, from the perspective of reject time, the hierarchical structure seems to be beneficial.
> In our CLBF (and existing LBFs such as Sandwiched LBF and PLBF), the Bloom filters used as components can be replaced with any classical data structure for approximate membership queries (e.g., Cuckoo filter and such hierarchical Bloom filters). Adaptively selecting and tuning this classical data structure according to the distribution of data and queries is a promising way to improve the performance of LBF.

---

### Official Review · Reviewer_HuJT · 2024-11-05

**Soundness:** 2
**Presentation:** 3
**Contribution:** 2
**Rating:** 3
**Confidence:** 3

**Summary:**

The paper proposes Cascaded Learned Bloom Filter (CLBF) to address the balance between the machine learning model size of learned Bloom filters and the Bloom filter size of classical Bloom filters.  The authors formulate an optimization approach based on dynamic programming to select configurations, with the reject time also part of the optimization objective. . Experiments using two benchmark datasets show that CLBF reduces memory usage by up to 24% and decreases reject time by up to 14 times compared to the state-of-the-art learned Bloom filter.

**Strengths:**

A cascading (learned) Bloom filter (LBF)  architecture that combines both sandwiched LBFs and partitioned LBFs (PLBFs), with classical BFs (as branch BFs).

**Weaknesses:**

The paper claims that the "optimization approach based on dynamic programming automatically selects configurations
that achieve an optimal balance between the model and filter sizes while minimizing reject time."  This claim is not substantiated at all.
You aim to optimize eq.(4), which seems to make sense, but which is largely meaningless.  First of all, unlike the classical BF where one can derive a low bound on the relation between false positive rate and filter size, what is the relation between the false positive rate of a LBF, $f^{(t})_i$ and the model size?    Second, given the cascading architecture, $\prod f^{(t})_i$ is not the overall false positive rate of the entire CLBF.  Most of section 3.3 makes little sense to me.

**Questions:**

See the weakness above.

1.  What is the relation between the false positive rate of a LBF, $f^{(t})_i$ and the model size?  Without an explicit expression, how do you optimize eq.(4) using dynamic programing?

2. Please justify that  $\prod f^{(t})_i$  yields the overall false positive rate of the entire CLBF.

3. In Section 3.3, You never establish the optimality of your dynamic programming formulation. Please formally establish the optimality of your solution.

4. How does the "depth" of your cascade, i.e., d, affect the overall balance between the model and filter sizes

---

> ### Author Response · Authors · 2024-11-23
>
> We appreciate the reviewer's insightful comments. We are happy to address the reviewer's concerns.
>
> > Weakness: The paper claims that the "optimization approach based on dynamic programming automatically selects configurations that achieve an optimal balance between the model and filter sizes while minimizing reject time." This claim is not substantiated at all.
>
> > Weakness: You aim to optimize eq.(4), which seems to make sense, but which is largely meaningless.
>
> The goal of our dynamic programming method is to select a configuration that minimizes the objective function defined in Equation (1). Equation (1) is a weighted sum of the memory size (the sum of the model size and filter size) and the reject time, and this is an objective function that directly expresses our goal of "achieving an optimal balance between the model and filter sizes while minimizing the reject time."
>
> Our dynamic programming algorithm, defined in Equation (4), can find the minimum of Equation (1) by obtaining the value of $\mathrm{dp}(1,1)$. By noting the following two points, it becomes clear that $\mathrm{dp}(1,1)$ is equivalent to the minimum value of Equation (1). Firstly, the constraint for $\mathrm{min}$ in equation (4), which is expressed as $\prod_{i=1}^{d-1} f^{(t)}_i = T$, is always satisfied when $d=1, T=1$, so this condition is no longer in effect when $d=1, T=1$. Secondly, the objective function for $\mathrm{min}$ in Equation (4) is the same as Equation (1) when $d=1$ (note that TBF_1 is located at the most upstream position in our proposed CLBF architecture, (Figure 2)). In other words, we can see that $\mathrm{dp}(1,1)$ expresses the minimum value of Equation (1) when $D$ and $f^{(t)}$ are optimally selected without any conditions, such as the total product of $f_i$. More precisely, our dynamic programming approach does not precisely find the exact minimum of Equation (1) because the thresholds $\theta$ are selected by testing several candidates, and the value $T$, which plays a key role in our dynamic programming, is discretized. However, our experimental results show that we can obtain good solutions using this dynamic programming.
>
> We have revised Section 3.3 to make this point clearer. The revised parts are highlighted in blue.

---

> ### Author Response · Authors · 2024-11-23
>
> > Weakness: First of all, unlike the classical BF where one can derive a low bound on the relation between false positive rate and filter size, what is the relation between the false positive rate of a LBF, $f^{(t)}_i$ and the model size?
>
> > Question 1: What is the relation between the false positive rate of a LBF, $f^{(t)}_i$ and the model size? Without an explicit expression, how do you optimize eq.(4) using dynamic programming?
>
> > Question 3: In Section 3.3, You never establish the optimality of your dynamic programming formulation. Please formally establish the optimality of your solution.
>
> In the case of our proposed CLBF architecture, the relationship between the false positive rate and the memory size can be explicitly expressed using $D$ and $f^{(t)}$. We give the following important insight in the first paragraph of Section 3.3: Once $D$ and $f^{(t)}$ are determined, the optimal values of $f^{(b)}$ and $f^{(f)}$ are determined immediately. This is derived from the conclusion obtained in the PLBF paper, which states that “when the ratio of keys and non-keys that enter a Bloom Filter ($\mathrm{BBF}$ or $\mathrm{FBF}$) is $g$ and $h$, respectively, the false positive rate of that Bloom Filter should basically be set to $Fg/h$” (where $F$ is the target false positive rate of CLBF). To clarify how the conclusion in the PLBF paper can be turned into our insight and to make it easier to understand how this insight is used in our dynamic programming method, we will give two examples below (please refer to Figure 2 to find the notation used in the explanation here at a glance).
>
> First, we will explain how the false positive rate of $\mathrm{BBF}_1$, i.e., $f^{(b)}_1$, is determined. We defined $g^{(b)}_1$ and $h^{(b)}_1$ as the ratio of keys and non-keys that are input to $\mathrm{BBF}_1$ when there is no filtering by $\mathrm{TBF}_1$. Then, when there is filtering by $\mathrm{TBF}_1$, whose false positive rate is $f^{(t)}_1$, the ratio of key and non-key input to $\mathrm{BBF}_1$ is $g^{(b)}_1$ and $f^{(t)}_1 h^{(b)}_1$, respectively. This is because the Bloom Filter has no false negatives, i.e., keys are not filtered by the $\mathrm{TBF}_1$, and only non-keys that pass through the $\mathrm{TBF}_1$ as false positives are input to the $\mathrm{BBF}_1$. In this case, the false positive rate of $\mathrm{BBF}_1$ $f^{(b)}_1$ can be expressed as $(F g^{(b)}_1) / (f^{(t)}_1 h^{(b)}_1)$.
>
> Next, we will explain how the false positive rate of $\mathrm{BBF}_2$, i.e., $f^{(b)}_2$, is determined. We defined $g^{(b)}_2$ and $h^{(b)}_2$ as the ratio of keys and non-keys that are input to $\mathrm{BBF}_2$ when there is no filtering by $\mathrm{TBF}_1$ and $\mathrm{TBF}_2$. Then, when there is filtering by $\mathrm{TBF}_1$ and $\mathrm{TBF}_2$, the ratio of key and non-key input to $\mathrm{BBF}_2$ becomes $g^{(b)}_2$ and $f^{(t)}_1 f^{(t)}_2 h^{(b)}_2$, respectively. This is because the Bloom Filter is a data structure that uses only a pure hash function without using the characteristics of the inputs, and it can be seen that the events where non-keys pass through $\mathrm{TBF}_1$ as false positives and the events where they pass through $\mathrm{TBF}_2$ as false positives are independent. In this case, the false positive rate of $\mathrm{BBF}_2$, $f^{(b)}_2$, can be expressed as $(F g^{(b)}_2) / (f^{(t)}_1 f^{(t)}_2 h^{(b)}_2)$. In other words, $f^{(b)}_2$ is determined by $f^{(t)}_1 f^{(t)}_2$.
>
> In this way, once $\prod_{i=1}^{d-1} f^{(t)}_i$ is determined, the false positive rate of $\mathrm{TBF}$ and $\mathrm{BBF}$ of depth $d$ or deeper can be determined without depending on the each value of $f^{(t)}_i (i=1,...,d-1)$ .
>
> Therefore, in the dynamic programming method defined in Equation (4), the value of the objective function is obtained by optimizing the structure below depth $d$ under the condition that $\prod_{i=1}^{d-1} f^{(t)}_i=T$.
>
> The minimum value of Equation (1) is obtained by recursively calculating $\mathrm{dp}$ and finally obtaining the value of $\mathrm{dp}(1,1)$. The optimality of our dynamic programming algorithm is guaranteed by the insight that, once $\prod_{i=1}^{d-1} f^{(t)}_i$ is determined, the false positive rate of the $\mathrm{FBF}$ and $\mathrm{BBF}$ of depth $d$ or deeper can be determined without regard to the value of each $f^{(t)}_i$.

---

> ### Author Response · Authors · 2024-11-23
>
> > Weakness: Second, given the cascading architecture, $\prod_{i} f^{(t)}_i$ is not the overall false positive rate of the entire CLBF. Most of section 3.3 makes little sense to me.
>
> > Question 2: Please justify that $\prod_{i} f^{(t)}_i$ yields the overall false positive rate of the entire CLBF.
>
> $\prod_{i} f^{(t)}_i$ is not the overall false positive rate of the entire CLBF, and we never wrote it as such. As explained in the answer to the previous question (and also explained in lines 260-263 of the paper), it is an important value for dynamic programming.
>
>
> > Question 4: How does the "depth" of your cascade, i.e., d, affect the overall balance between the model and filter sizes
>
> The "depth" (denoted $D$ in the paper) in CLBF is a value determined by optimization, so the CLBF users do not have to adjust $D$ themselves. On the other hand, the following two parameters are adjustable parameters:
>
> 1. The number of weak learners given when constructing LBFs, such as PLBF or Sandwiched LBF.
> 2. The number of weak learners given when constructing CLBF, i.e., $\bar{D}$. This is a value determined by the user of CLBF (only $D$ of the $\bar{D}$ weak learners are used by CLBF).
>
> Therefore, we conducted the following two experiments:
>
> 1. The effect of $D$ on the memory balance of PLBF.
> 2. The effect of $\bar{D}$ on the memory balance of CLBF.
>
> The results are shown in Appendix B. The results can be summarized as follows:
>
> 1. As $D$ increases, the memory usage of ML Model in the PLBF, of course, increases, while the memory usage of the backup Bloom Filter used by the PLBF tends to decrease (Figure 9). This is because the accuracy of the ML Model improves as the ML Model becomes larger, and it becomes possible to achieve the target false positive rate with a smaller backup Bloom filter. The total memory usage (the sum of the memory usage of the ML Model and the Bloom Filter) tends to decrease when $D$ is small, and tends to increase when $D$ is large. The optimal $D$ varies depending on the dataset and the target false positive rate, but in all cases, it can be seen that CLBF is able to select a $D$ that is almost optimal.
> 2. In the case of CLBF, when $\bar{D}$ is small, increasing $\bar{D}$ monotonically decreases the total memory usage and monotonically increases the size of the machine learning model used (Figure 8). On the other hand, if we increase $\bar{D}$ beyond a certain value, the overall memory usage and the size of the ML Model will no longer change. This is because CLBF selects the value of $D (\in \{1, 2, \dots, \bar{D}\})$ that minimizes the total memory usage.

---

### Meta-Review · Area_Chair_6FRX · 2024-12-24

**Metareview:**

The paper proposes a Cascaded Learned Bloom Filter (CLBF) to optimize the trade-off between memory usage and reject time in Bloom filter structures, using a dynamic programming approach for configuration selection. The proposed method demonstrates improved memory efficiency and reject time compared to prior methods, with experiments showcasing its benefits on benchmark datasets.

Reviewers acknowledged the relevance of optimizing Bloom filters for modern applications but raised critical concerns about the paper’s novelty, computational overhead, and applicability. While the proposed cascade structure and dynamic programming method provide an innovative perspective, the approach was perceived as a generalization of existing techniques, offering incremental contributions rather than fundamentally novel insights. Additional concerns included the scalability of the method for real-world applications, the limited exploration of trade-offs, and the lack of clarity in some mathematical formulations. The authors provided detailed rebuttals, including additional experiments and revisions, but these responses did not fully resolve the reviewers’ concerns.

Considering these factors, the AC recommends rejection. Despite its potential, the paper does not sufficiently demonstrate a significant leap in innovation or address practical limitations to justify acceptance in its current form.

**Additional Comments On Reviewer Discussion:**

Please refer to the meta review.

---

### Decision · Program_Chairs · 2025-01-22

Reject